🔓 | **Open Peer Review** | Environmental Microbiology | Research Article

# Responses of rhizosphere bacterial communities with different niche breadths to liquid fertilizer produced from *Fuji* apple wastes during planting process

Fengan Jia,[1] Fan Chang,[1] Qingan Jia,[2] Yan Li,[1] Lisha Zhen,[1] Rui Lv,[1] Yun Xie[3]

**ABSTRACT** The application of fertilizer in agroecosystems exerts a significant influence on the regulation of soil microbial community and soil properties. However, the impact of liquid fertilizer derived from apple wastes on soil properties, microbiota, and plant growth remains unclear. In this study, we investigated the effects of apple waste liquid fertilizer on rhizosphere microbiota and soil characteristics by analyzing both the bacterial community and physicochemical properties of the soil. The results demonstrated that the application of liquid fertilizer significantly influenced the diversity and structure of bacterial communities. Furthermore, the application mainly increased the niche breadth of bacterial communities, while exerting minimal impact on the assembly process. The rhizosphere bacterial communities treated with a high concentration of liquid fertilizer exhibited increased diversity and broader niche during the late planting period. Moreover, bacterial cooccurrence networks revealed that communities following the application of liquid fertilizer exhibited reduced network nodes and edges, as well as smaller network sizes. These communities were characterized by increased modularity and interconnectedness facilitated by key bacterial indicators such as *Sphingomonas*. The presence of indicators representing different niche breadths in bacteria showed significant correlations with available nitrogen (AN) and available phosphorus (AP), while positively correlating with the Bray-Curtis dissimilarity. Following the application of high concentration liquid fertilizer, the number and relative abundance of specialist ASVs such as *Sphingomonas*, Gp6, and *Arthrobacter* increased. The results suggested that liquid fertilizer impacted the soil AN and AP through alterations in the recruitment pattern of specialist ASVs. Also, liquid fertilizer treatment significantly increased the abundance of carbohydrate metabolism and biosynthesis of other secondary metabolites pathway genes of bacterial communities. Plant height, leaf length, leaf width, and dry weight of plants were significantly increased in plants treated with liquid fertilizer These results suggested that the application of liquid fertilizer could result in variations in the microbial community composition, structure, and function. The composition of the liquid fertilizer and the potential functional microorganisms still need to be further studied.

**IMPORTANCE** The issue of arable soil degradation is of global importance, and the utilization of agricultural waste plays a pivotal role in enhancing soil properties and promoting sustainable crop production. In particular, elucidating how agricultural waste affects the microbial composition and regulates soil properties is extremely relevant at both fundamental and practical levels. The bioinformatics analysis of microbiota from field experiments revealed that the application of liquid fertilizer from apple wastes significantly affected the structure and co-occurrence network of rhizosphere bacterial communities. Also, it was observed that the planting time also influenced the community structure. Additionally, the soil available nitrogen (AN) and available phosphorus

Address correspondence to Fan Chang, fox387@163.com.

The authors declare no conflict of interest.

See the funding table on p. 17.

(AP) were significantly influenced by bacterial communities, leading to alterations in soil properties. Finally, liquid fertilizer treatment significantly affected the function of soil bacteria community and promoted the growth of bok choy.

**KEYWORDS** apple wastes liquid fertilizer, rhizosphere bacterial community, soil properties

Interactions between the soil microbial communities and soil properties have an important role in driving soil functions in agro-ecosystems (1). The microbial community in the soil regulates nutrient, physiological, and biochemical properties of soil in farmland (2). The microbial-mediated alterations in soil properties have the potential to impact local microbiome establishment and activity, while also enhancing nutrient utilization through biological weathering, thereby promoting the overall health of both soil and microbiota (3). Microorganisms play a crucial role in regulating the carbon and nutrient cycles of soil (4), enhancing soil structure (5), suppressing plant diseases and insect pests (6), as well as other vital biological processes. Moreover, they serve as sensitive indicators of soil health, providing valuable insights regarding their condition (7, 8). At the same time, the rhizosphere microbiota simultaneously play a pivotal role in the nutrient transformation of plant roots (9). Numerous studies have demonstrated that the diversity and succession of rhizosphere microbial communities are indispensable for enhancing individual plant and ecosystem functions (10, 11).

The issue of farmland soil degradation cannot be overlooked due to the growing population, climate change, excessive soil burden, and continuous cultivation (12). The improvement of soil nutrients is a crucial measure for maintaining soil health, enhancing crop yields, and safeguarding the environment (13, 14). Among these approaches, agricultural waste application plays a significant role in enhancing soil properties and promoting sustainable crop production. Numerous strategies have been devised to enhance soil quality through the utilization of natural substances. Mineral amendments are used to immobilize heavy metals, regulate soil pH, and reduce the bioavailability of extractable toxic compounds (15). Appropriate microorganisms, plant growth promoters, and other biological amendments are used to resist pathogens and improve plant growth (16). Additionally, organic amendments like fruit residue, straw, animal manure, and other agricultural wastes are utilized to enhance soil structure, increase organic matter content, and stimulate microbial activity (17, 18).

The utilization of fermented fruit wastes as soil amendments and fertilizers in agriculture and ecosystems has garnered significant attention. Fruit wastes are abundant in organic matter and cellulose, which can enrich soil beneficial microorganisms and promote plant growth (19). The organic matter present in the fruit wastes reacts with the salt in the soil to form a stable complex, thereby mitigating the toxicity of salt toward plants (20). Findings demonstrate a significant improvement in both crop yield and soil properties following the application of fruit waste fertilizers (21, 22). At the same time, fruit waste fertilizers can provide nutrients and promote the diversity of microbial communities. Studies have found that organic acids and other components contribute to the maintenance of soil water retention, permeability and fertility, and functioning of soil microbiome ecosystems (23, 24).

The Food and Agriculture Organization of the United Nations has officially designated Shaanxi Province in China as the world's premier apple cultivation region, renowned for its quality and productivity. Moreover, it holds the distinction of being the largest contiguous area dedicated to apple farming worldwide (25). The management of fallen apples has always posed a challenging issue in the advancement of the apple industry in Shaanxi. Here, we obtained liquid fertilizer from fallen apples through anaerobic fermentation and conducted field experiments to investigate the impact of different concentrations of the fertilizer on soil and vegetable rhizosphere microhabitats' bacterial structure, diversity, and community succession over time. We constructed the correlation networks of soil microbiota with varying niche breadths and examined the impact of

liquid fertilizer concentration on bacterial communities, as well as its interaction with soil physiological and biochemical properties.

## MATERIALS AND METHODS

### Field experiment and pretreatment

The experimental plot was located at the experimental station of the Shaanxi Agricultural Machinery Research Institute, Xianyang (latitude 34°21′7″N, longitude 108°42′34″E, elevation 395 m), Shaanxi province. The mean annual temperature is 9.0–13.2 °C, the average annual sunshine is about 2,200 h, the annual precipitation is 537–650 mm, and the soil type is cinnamon soil without agricultural cultivation. The major soil parameters were as follows: organic matter (OM) 27.63 g/kg, total nitrogen (TN) 1.27 g/kg, available nitrogen (AN) 84.19 mg/kg, total phosphorus (TP) 0.94 g/kg, available phosphorus (AP) 33.37 mg/kg, total potassium (TK) 19.80 g/kg, available potassium (AK) 379.44 mg/kg, pH 8.57, and electrical conductivity (EC) 203.94 µS/cm.

The liquid fertilizer was produced from apple waste through an anaerobic mesophilic process. The average temperature in the fermentation reactor was maintained at 38°C ± 1°C, with a holding time of 15 days and a total fermentation period of 60 days. After fermentation, a 200 micron filter was used to obtain the filtrate, which served as the stock for the apple waste liquid fertilizer. The liquid fertilizer had a pH of 4.5, soluble solids of 5.5°Bx, total acid concentration of 3.5 g/L (including 2.4 g/L malic acid). Finally, the liquid fertilizer was stored in closed plastic barrels at a temperature of 4°C.

The interspersed trials were established in a randomized block design (CRD) with three treatments replicated three times, and each replicate plot (5 × 5 m) was spaced 2 m apart. In each block, a total of 300 bok choy (*Brassica rapa*) individuals were planted in 20 rows and 15 columns, and the plant spacing was 20 cm, with a row spacing of 30 cm. The experimental design included three treatments: CK (control experiment, unfertilized), L (low concentration apple waste liquid fertilizer, diluted 400 times), and H (high concentration apple waste liquid fertilizer, diluted 200 times). Each experimental plot received an application of 5 kg of liquid fertilizer, while the soil moisture content was adjusted to a range of 40%–50% (wt/wt) (Fig. 1).

### Soil samples collection

Rhizosphere soil samples were collected at the beginning of May 2023. The samples were collected on day 2, day 16, and day 32 after treatments. Six samples were collected randomly from each plot. For each sampling spot, 10–15 healthy plants with consistent growth were randomly selected, vigorously shaken by vortex oscillation, and the soil adhered to the roots was collected as one rhizosphere soil sample. Then, all collected rhizosphere soil samples were sieved through a 2 mm sieve and divided into sub-samples for analysis of microbial community diversity, physicochemical properties, and a backup. The samples in the same group use the same sieve, which had been sterilized at 121°C for 20 min before use. In total, 54 samples (6 samples × 3 treatments × 3 time points) were collected, immediately frozen in liquid nitrogen, and stored at −80°C for further analysis.

### DNA extraction and 16S rDNA sequencing

Genomic DNA was extracted from 0.5 g soil using the Fast DNA SPIN Kit for Soil (MP Biomedicals, USA). The quality of the extracted DNA was checked using an Agilent 2100 Bioanalyzer (Agilent Technologies, USA). Total DNA was eluted in 50 µL of Elution buffer and stored at −80°C until PCR. The V3–V4 regions of the bacterial 16S rRNA gene were amplified using the primers 341F (5′-CCTACGGGNGGCWGCAG-3′) and 805R (5′-GAC-TACHVGGGTATCTAATCC-3′) (26). The next-generation sequencing (NGS) library preparations and sequencing were performed by LC-Bio Technology Co., Ltd., Hang Zhou, China. DNA samples were quantified using a Qubit 2.0 fluorometer (Invitrogen, Carlsbad, CA,

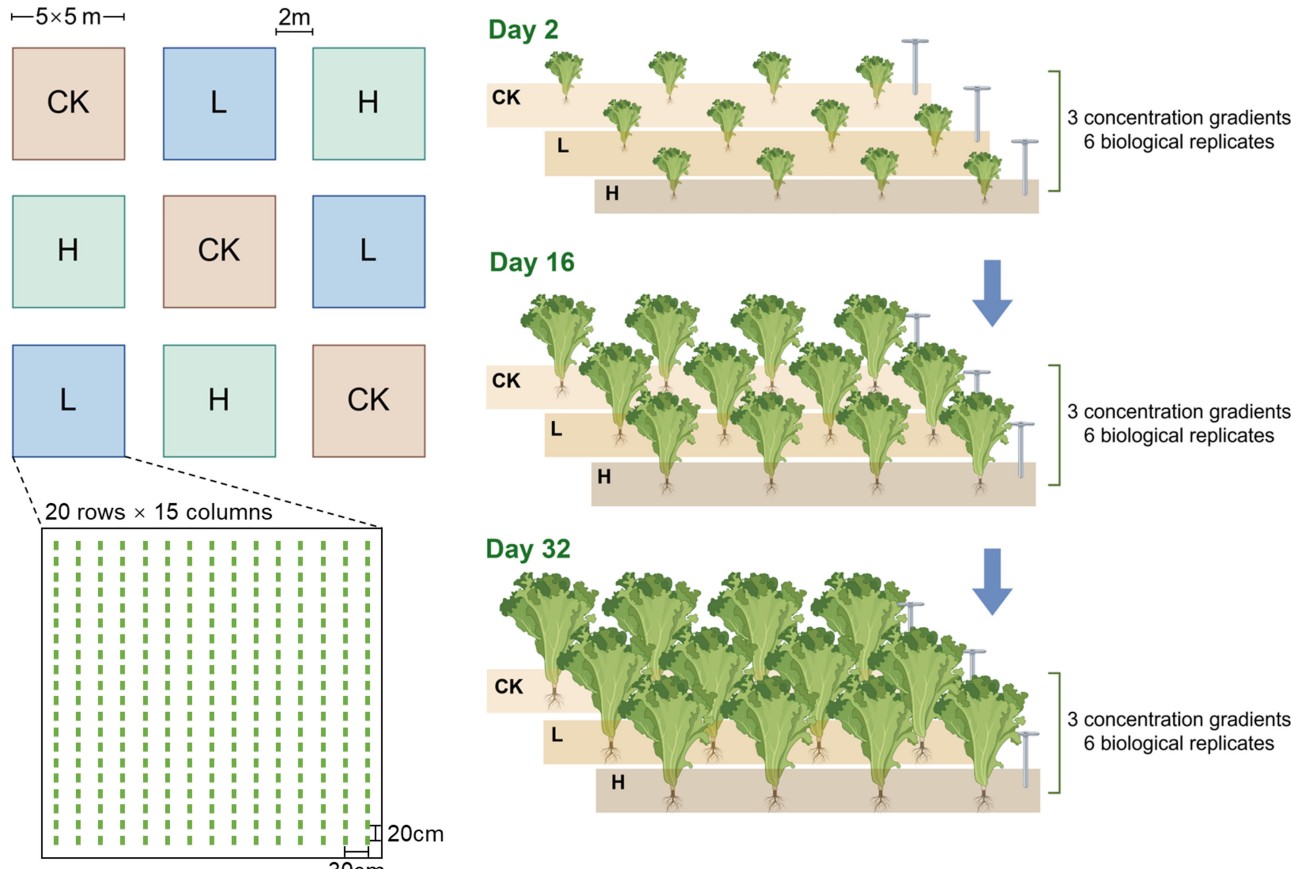

**FIG 1** Pictorial representation of the bok choy planting and sampling process in the experimental plot.

USA). The libraries were generated with the NEBNext UltraTM DNA Library Prep Kit for Illumina (New England Biolabs, USA) and analyzed by the Illumina platform, according to the manufacturer's protocol.

The raw sequences were processed using Usearch v10 (https://www.drive5.com/usearch/manual10/) to perform a quality control, ASV table generation, and bioinformatics analysis (27). Briefly, the forward and reverse sequences were joined and assigned to samples based on the barcode sequences and then truncated by removing the barcode and primer sequences. Quality filtering on joined sequences was performed, and sequences with length <200 bp, ambiguous bases, and expected error per base rate >0.01 were discarded. Subsequently, sequences were dereplicated, and singletons (with a minuniquesize <8) were removed. Pre-processed sequences were clustered into amplicon sequence variants (ASVs) using the unoise3 denoising algorithm, and chimeric sequences were simultaneously removed (28, 29). The final effective sequences were obtained for further analysis. The *sintax* algorithm of Usearch10 recognized taxonomic classification of ASVs against the Ribosomal Database Program v11.4 (RDP, https://ngdc.cncb.ac.cn/databasecommons/database/id/237 and https://www.hsls.pitt.edu/obrc/index.php?page=URL1100791247), preclustered at 97% sequence identity, and the confidence threshold for the RDP classifier for ASVs is 0.8 (30). The raw data of sequences were available at the National Center for Biotechnology Information (NCBI) under BioProject ID PRJNA1138972.

## Soil physicochemical properties and plant parameters analysis

To determine the physicochemical properties of soil, samples were taken from the top layer at 0–20 cm depth from six random locations at each site for each experimental

plot. Soil samples at the full depth of the profile were evenly mixed, sealed in sterile sampling bags, and transported to the laboratory on ice, where they were air-dried and sieved to prior use. Soil organic matter (OM) was measured by dichromate oxidation and ferrous sulfate titration according to the Walkley-Black method (31). Soil total nitrogen (TN) was determined using the Kjeldahl method (32). Available nitrogen (AN) was measured via the NaOH diffusion method (33). Soil total phosphorus (TP) was measured using the Mo-Sb anti-spectrophotometric method (34). Available phosphorus (AP) was determined using the molybdenum blue method (35). Soil total potassium (TK) and available potassium (AK) were measured by flame photometry after extraction with sodium hydroxide and ammonium acetate, respectively (36). Dried samples were mixed and shaken with deionized water at a volume ratio of 1:2.5, followed by centrifugation at 12,000 rpm for 5 min to determine the pH and electrical conductivity (EC) using the FE28 desktop pH and conductivity meter (Mettler Toledo, USA) (37).

In order to evaluate the effect of liquid fertilizer on the plant morphological and physiological characteristics, parameters including plant height (cm), number of leaves, maximum leaf length (cm), maximum leaf width (cm), root length (cm), and dry weight (g) of randomly selected 10 healthy plants were determined after 32 days of treatments.

## Statistical analysis

Statistical analysis and graphic display of data were performed using R software version 4.1.2 (38). Alpha diversity was assessed with the Chao1 index, Shannon index (vegan package, version 2.5–6) (39), and Faith's phylogenetic diversity (PD) index (picante package version 1.8.2) (40). Phylogenetic tree construction was conducted using the Usearch10 *cluster_agg*. Statistical significance in alpha diversity between treatments and time points was calculated with one-way analysis of variance (ANOVA) and Tukey's multiple comparison tests. Beta diversity was evaluated by principal coordinate analysis (PCoA) based on the Bray-Curtis distance. The significance of the difference in community dissimilarity between treatments and time points was tested with a PERMANOVA by the *adonis* function in the vegan package.

To compare the assembly processes and multifunctionality of rhizosphere microbiota, beta nearest taxon (βNTI) index and niche breadth in different treatments and time points were calculated. The βNTI was calculated using the picante package (version 1.8.2) to investigate community assembly processes and quantify phylogenetic structure (41). Niche breadth index was calculated using Levin's niche breadth equation with the spaa package (version 0.2.1) (42). Classification of ASVs in specialist/generalist/opportunist species was based on the deviation of niche breadth index by EcolUtils package (version 0.1). The ASVs were categorized as specialist species, generalist species, and opportunist species based on their niche breadth index falling below the lower limit, exceeding the upper limit, and within the 95% confidence interval, respectively (43). Functional assignments were predicted using the Tax4Fun2 package (version 1.1.5) (https://github.com/fjossandon/Tax4Fun2) based on the ASVs with the Ref99NR database (44).

To further characterize the impact of liquid fertilizer on rhizosphere microbiota, we evaluated the composition and co-occurrence patterns of bacterial communities of different niche breadth. "Indicspecies" package (version 1.7.9) was used to identify potential ASV indicators of each group (CK, L, and H) (45). Co-occurrence networks of indicators were constructed using the "igraph" package (version 1.2.5). A valid co-occurrence was considered a statistically robust correlation between ASV indicators when the correlation coefficient was >0.8 or < −0.8 and the *P* value was below 0.01. The co-occurrence network graph was visualized using Gephi (version 0.9.2) (46). The three centrality indices PageRanks, Eccentricity, and Betweeness Centrality of the network were calculated by Gephi, and the "centrality" of indicators was defined by considering the intersection of these indexes.

To investigate the interrelationships among soil properties and bacterial communities of different niche breadth, Spearman's rank correlation analysis was employed

to evaluate the associations between these variable sets. Prior to correlation analysis, the "Hmisc" package (https://cran.r-project.org/package=Hmisc) was used to eliminate collinear factors of soil properties. The factors were transformed and normalized to evaluate their association with ASVs of different niche breadth, followed by the application of Mantel tests (47).

## RESULTS

### Impact of liquid fertilizer concentration on the bacterial community and structure in the rhizosphere

To determine the effects of different concentrations of liquid fertilizer on the microbiota in soil, alpha and beta diversity were calculated to compare the bacterial community of samples over time and between different treatments. The Chao1 index and Faith's PD index of rhizosphere bacterial communities showed a significant increase with fertilizer concentration, and this increase was significantly influenced by time. The Shannon index followed the same trend although there were no significant differences observed (Fig. 2A; Table S1). To gain insight into similarities in the bacterial community structures between the different concentrations and time groups, PCoA of beta diversity analysis was performed based on the Bray-Curtis distances, which demonstrated distinct community structures observed among the CK, L, and H groups, with the variation in structure being associated with time (Fig. 2B). A clear segregation in community structures was exhibited between the CK, L, and H groups on Day 2 and Day 32 in the horizontal coordinate (PC1 axis), with the first two principal components representing 14% and 11% of the total variations. PERMANOVA testing confirmed that a significant separation occurred between the groups ($R^2 = 0.27$, $P = 0.001$) (Fig. 2B). Next, the Bray-Curtis dissimilarity was calculated to examine the changes in microbiota composition across different treatments over time. The Bray-Curtis dissimilarity of group H showed a significant decrease compared to the other two groups, primarily due to a notable reduction on Day 32 (Fig. 2C; Table S2). In addition, the bacterial communities in the CK group showed significant fluctuations at different timepoints. There was an initial increase followed by a decrease in alpha diversity, beta diversity distance, and Bray-Curtis similarity (Bray-Curtis similarity = 1 – Bray-Curtis dissimilarity, Fig. 2).

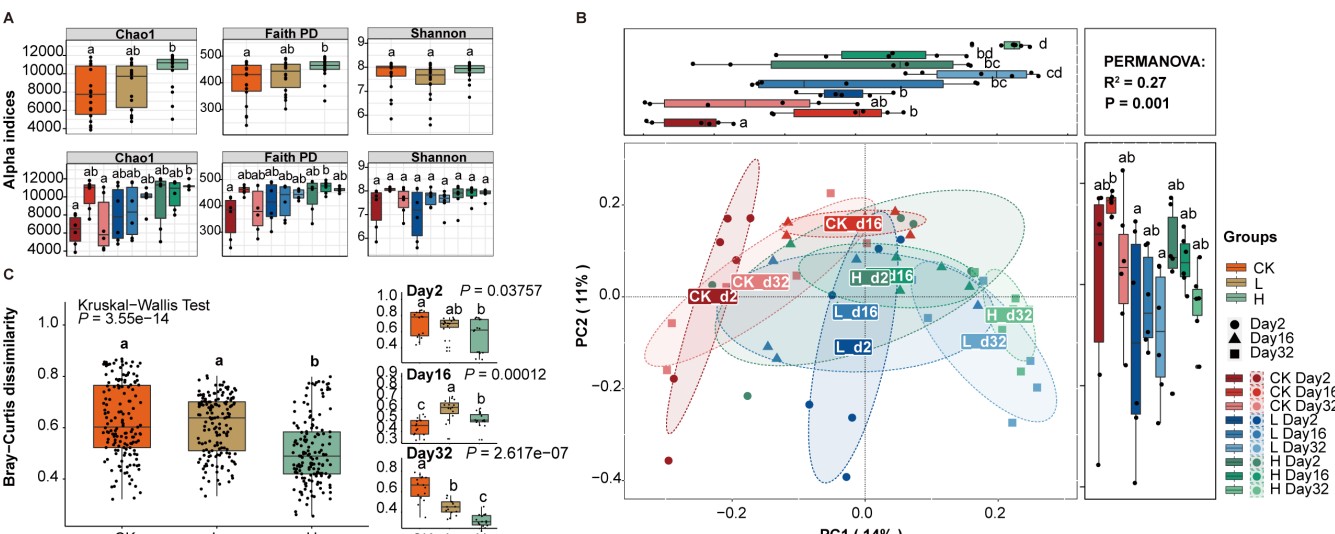

**FIG 2** Microbial diversity and structure of the bacterial microbiota in liquid fertilizer with different concentrations at three time points. (A) Chao1, Faith's PD, and Shannon indices of bacterial community. (B) The beta diversity assessed by principal coordinates analysis (PCoA, pairwise comparisons based on PERMANOVA). (C) Bray–Curtis dissimilarity in different concentrations and changes at three time points. ANOVA and Tukey's multiple comparison tests for each group, and the statistical significances ($P < 0.05$) were indicated by different letters. CK, unfertilized; L, low concentration; H, high concentration.

## Impact of liquid fertilizer concentration on the community assembly pattern and niche breadth of rhizosphere microbiota

βNTI values provided insights into the potential roles of deterministic and stochastic effects in bacterial community dynamics. The community assembly in different groups was primarily dominated by stochastic processes (βNTI > 2). While the βNTI distribution of the H group gradually shifted along time, from stochastic process (βNTI > 2) to deterministic process (βNTI < 2), which was not observed in the other two groups (Fig. 3A; Table S2). It was also observed that high treatment concentrations led to a significant increase in niche breadth, and this change was found to be time-dependent. And the proportion of high niche breadth increased with treatment concentration. Specifically, on Day 32, both the L and H groups exhibited a noteworthy increase in the high value of niche breadth (Fig. 3B; Table S2). Besides, the βNTI and niche breadth in the CK group showed a fluctuation, initially increasing and then decreasing.

## Genus with different niche breadth showed different responses to the concentration and time

The ASVs were categorized into three groups based on their niche breadth. It was observed that the relative abundance of specialist species was higher (from 52.84% to 69.49%), followed by opportunist species (from 25.13% to 38.89%). At the genus level, specialist species (6 genera) had the highest number of differential species. *Sphingomonas* (1.96% to 5.63%) and *Lysobacter* (0.49% to 2.17%) exhibited significant increases in relative abundance as fertilizer concentration and planting time increased, whereas *Gp6* (4.51% to 1.70%), *Gemmata* (1.90% to 0.35%), and *Pirellula* (1.60% to 0.40%) showed the opposite trend. While, the L group showed a higher relative abundance of *Pseudoxanthomonas* (1.31%), *Bacillus* (2.31%), *Ohtaekwangia* (1.36%), and *Devosia* (0.94%). In particular, opportunist species *Sphingomonas* (0.19%–0.46%) and *Bacillus* (0.22%–0.46%) showed consistent trends with the same genus in the specialist species (Fig. 4; Table S3).

## The composition of bacterial communities and co-occurrence networks of varying niche breadth in different treatment concentrations

Inter-group indicators were screened to make a co-occurrence network, and three species groups of generalist, opportunist, and specialist were labeled, respectively (Fig. 5). The application of liquid fertilizer was found to have a significant impact on the network characteristics of bacterial communities in soil. The number of nodes (1,075) in the CK group was nearly twice as many as that in the liquid fertilizer groups (655 in L and 595 in H group), and the number of edges (16,100) was over 10 times greater than that in the L (1,418) and H (1,229) groups. The average degree distribution of CK was seven times higher than that of groups L and H (29.95 vs 4.33 and 4.13), while the average clustering coefficient only slightly surpasses the two groups (0.46 vs 0.32 and 0.33). Additionally, group CK exhibits the smallest average path length among these three groups (3.92 vs 6.32 and 6.86). The network characteristics of the CK group suggested that its co-occurrence network of indicator communities exhibited small-world property. On the other hand, the modularity of the liquid fertilizer groups was higher than that of the CK group, with the H group showing the highest value (0.496, 0.698, and 0.742 in CK, L, and H group). This suggested that the network structure in the H group had the densest connections between nodes. Additionally, the H group had the highest number of modules (17 vs 82 and 84), while the number of max K-Core was 35 in the CK group, and 8 in the L and H groups. These findings indicated that there were closer relationships between microbiota in groups L and H co-occurrence networks, and different indicators were inter-connected through certain key bacterial nodes (Fig. 5; Table S4). Furthermore, despite CK having a greater number of network nodes and edges, as well as a larger overall network size, there was no disparity in the number of central indicators identified by the centrality indices (133, 127, and 101 in CK, L, and H group).

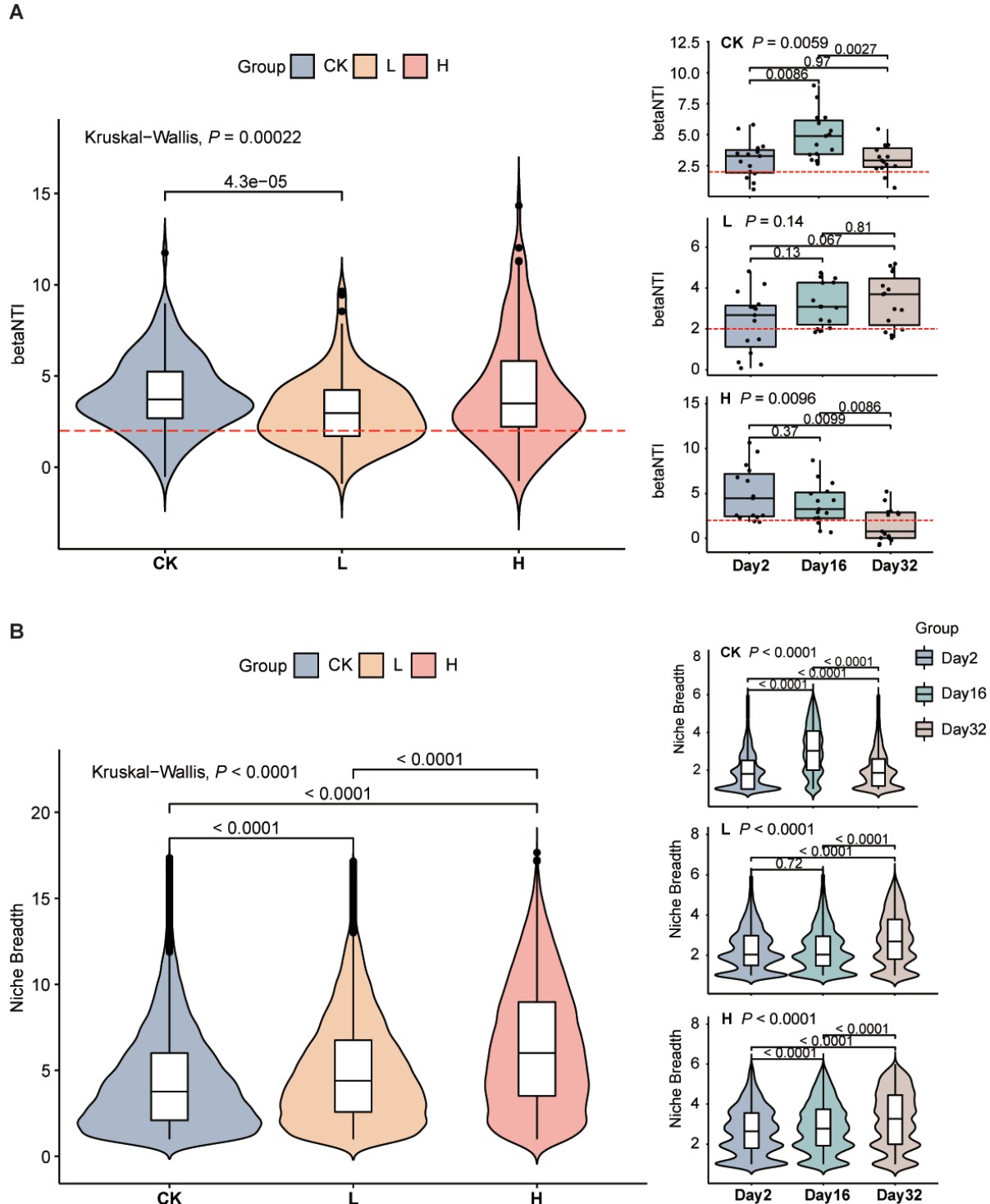

**FIG 3** Differences in beta nearest taxon index (βNTI) and niche breadth of rhizosphere bacterial communities under different liquid fertilizer treatments at three timepoints. (A) betaNTI comparison. (B) Niche breadth comparison. Kruskal-Wallis (K-W) and Wilcoxon tests were used for comparisons, and significant differences were indicated by horizontal bars with corresponding *P* values. CK, unfertilized; L, low concentration; H, high concentration.

Among the three species groups characterized by niche breadth, the central indicators mainly consisted of specialist species. The relative abundance of central indicators in group CK was low; the central indicator among the top 15 high-abundance ASVs in the opportunist cluster was only *Terrimonas* (ASV_301). While the treatment of liquid fertilizer was found to increase the number of central indicators in high-abundance ASVs.

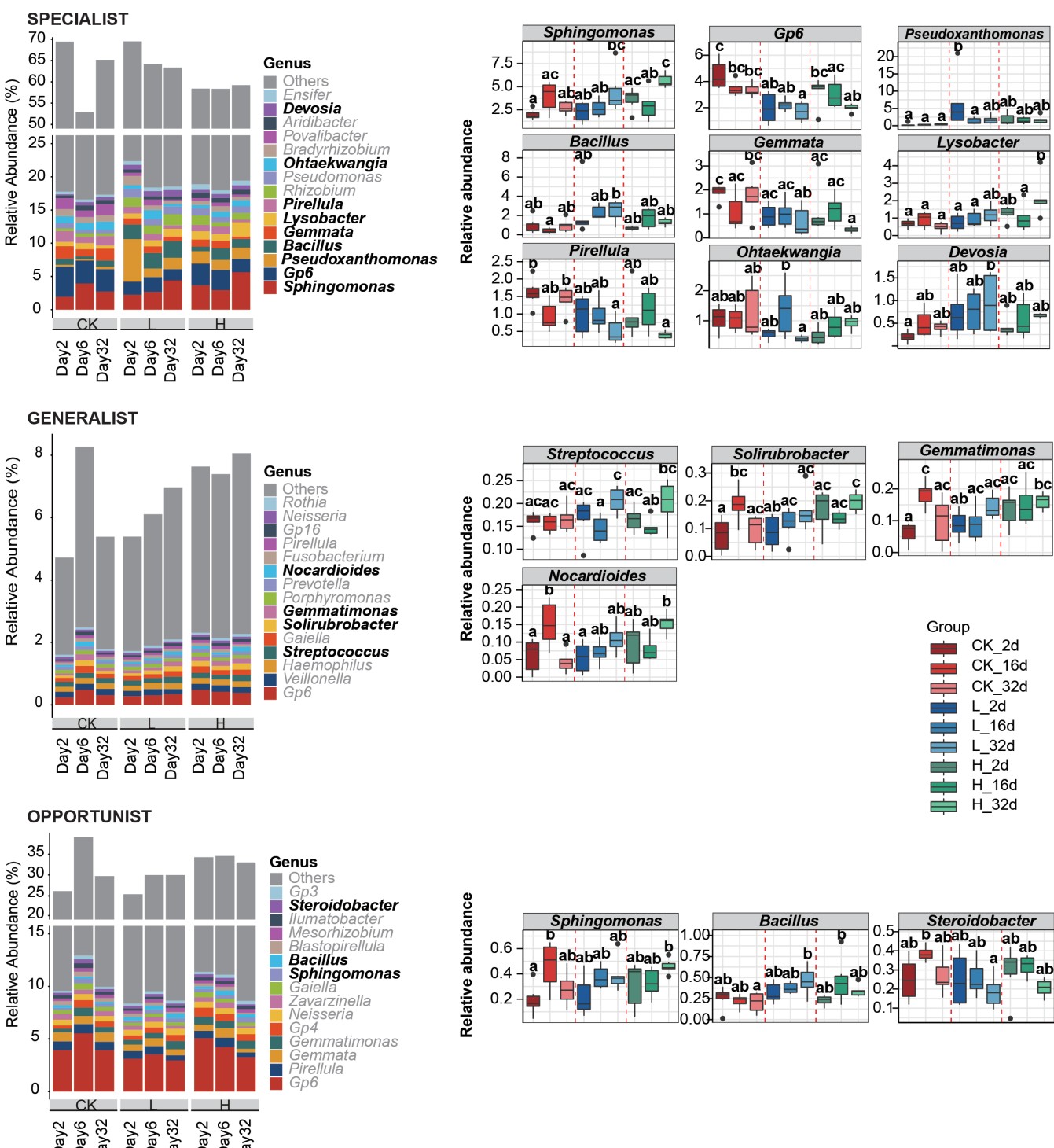

**FIG 4** Comparison of relative abundance of bacterial genera with different niche breadth among the CK, L, and H groups at three timepoints. Genera abundance with significant differences was represented in black bold. ANOVA and Tukey's multiple comparison tests for each group and the statistical significances ($P < 0.05$) were indicated by different letters. CK, unfertilized; L, low concentration; H, high concentration.

In group L, the central indicators consisted of 12 high-abundance ASVs, including *Sphingomonas* (ASV_22, 37, 14), *Rhodococcus* (ASV_52), *Micromonospora* (ASV_362), and *Aridibacter* (ASV_64) in the specialist, Gp16 (ASV_1295) and *Gemmatimonas* (ASV_3044) in the generalist, and *Morococcus* (ASV_362), *Subdivision3* (ASV_248), and *Gemmatimonas* (ASV_378) in the opportunist (Fig. 5B). In group H, the central indicators consisted of 15

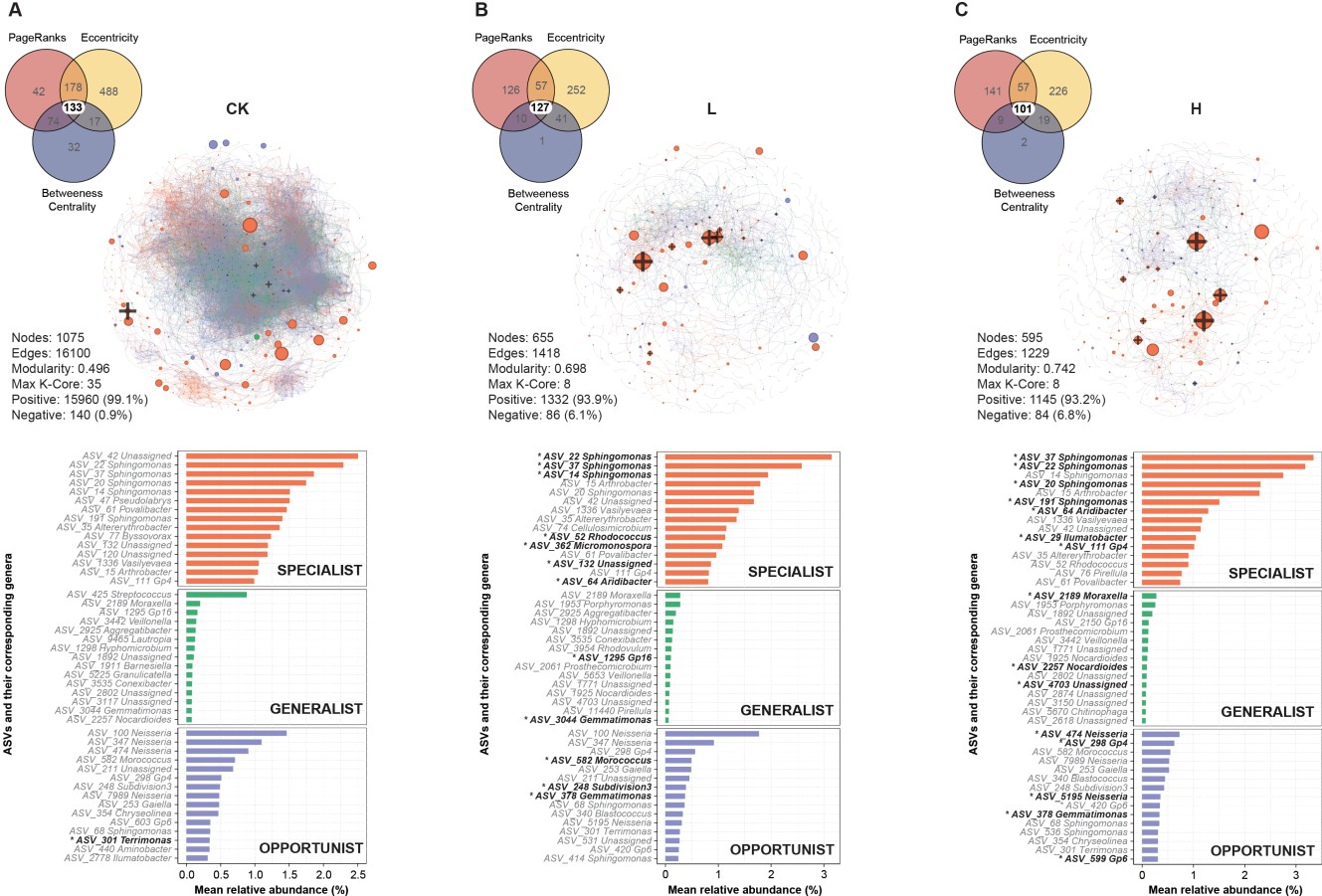

**FIG 5** Correlation network of ASV indicators and centrality of different niche breadth among CK (A), L (B), and H (C) groups. The Venn diagram illustrated the distribution of PageRanks, Eccentricity, and Betweeness Centrality for the three centrality indices of the indicators. The numbers on a black and bold white background indicated the centrality ASV indicators. The network diagram displayed specialist, generalist, and opportunist ASVs in vibrant orange, refreshing green, and deep navy colors respectively. Additionally, the bold black "+" symbolizes the central indicators based on their centrality indices. The bar chart showed the top 15 indicators of various niche breadth indicators in different co-occurrence network groups. Black bold ASVs with asterisks indicated central indicators. CK, unfertilized; L, low concentration; H, high concentration.

high-abundance ASVs, including *Sphingomonas* (ASV_37, 22, 20, 191), *Aridibacter* (ASV_64), *Ilumatobacter* (ASV_29), and Gp4 (ASV_64) in the specialist, *Moraxella* (ASV_2189) and *Nocardioides* (ASV_2257) in the generalist, and *Neisseria* (ASV_474), Gp4 (ASV_298), *Neisseria* (ASV_5195), *Gemmatimonas* (ASV_378), and Gp6 (ASV_599) in the opportunist (Fig. 5C). In particular, ASVs belonging to *Sphingomonas* were not only the most abundant species but also served as central indicators in the liquid fertilizer treatment groups.

## The correlation between indicators of various niche breadth and soil properties

The intersection of indicators from the co-occurrence network of groups CK, L, and H was identified. Three subgroups of rhizosphere bacterial communities based on the niche breadth were defined. Subsequently, the correlation between the subgroups and soil properties was then investigated. Overall, there was a significant correlation between soil SOM, N, and K. While soil K and P had significant correlation with soil heavy metal ions ($P < 0.01$). In particular, TK was positively correlated with SOM, TN, AN, TP, Cu, Fe, and Mn but negatively correlated with AK and AP. This result was the opposite of AK (Fig. 6A). The physicochemical properties of soil were in Table S7.

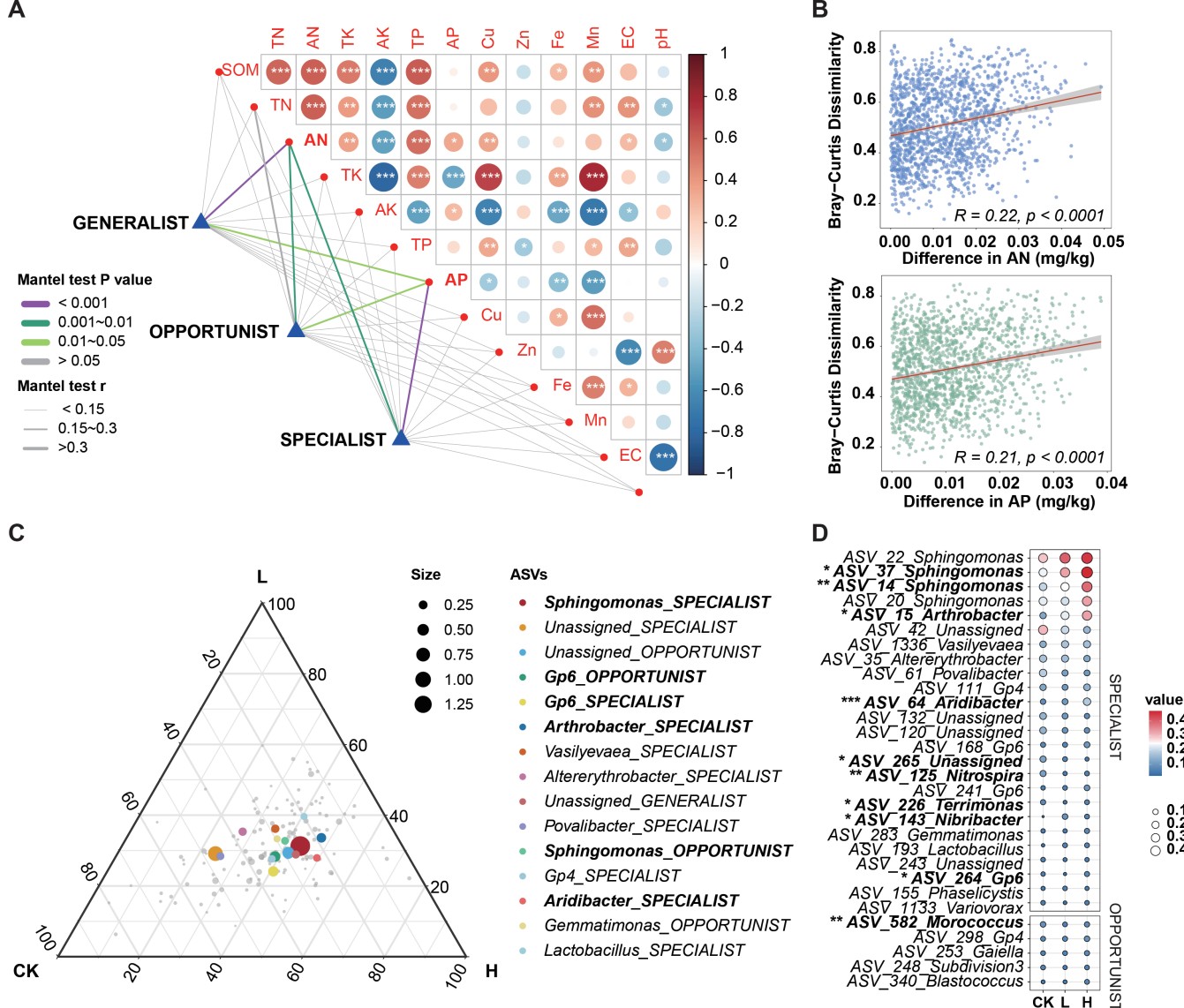

**FIG 6** Relationships between soil properties and subgroups of centrality indicator ASVs. (A) Correlation analysis of soil properties and specialist, generalist, and opportunist ASVs. (B) Linear regression analysis of correlations between Bray–Curtis dissimilarity of ASVs and contents of AN and AP. (C) The relative abundance distribution of the top 15 genera in groups CK, L, and H. The bold represented the genera with significant differences in the D chart. Each genus was marked with its own niche breadth index, connected by "_". (D) The difference of the top 30 ASVs in relative abundance among CK, L, and H groups. Each ASV was marked with its own genus, connected by "_". ANOVA and Tukey's multiple comparison tests for each group. Statistically significant differences were indicated by asterisks and bold font (*$P < 0.05$; **$P < 0.01$; ***$P < 0.001$). CK, unfertilized; L, low concentration; H, high concentration.

The Mantel test was employed to investigate the correlation between indicators of diverse niche breadth and soil properties. The results showed that indicators were significantly correlated with soil AN and AP. Additionally, compared to the opportunist subgroup, the specialist subgroup and the generalist subgroup exhibited a stronger correlation with AN and AP in soil. (Mantel's $r > 0.3$, $P$-value $< 0.01$, Fig. 6A). Notably, the Bray-Curtis dissimilarity of indicators showed a significant positive correlation with AN and AP content (Fig. 6B).

The bacterial genera distribution, relative abundance, and corresponding ASVs analysis revealed that *Sphingomonas* exhibited the highest abundance, followed by Gp6. The relative abundance of *Sphingomonas*-associated ASVs_37 and ASVs_14 exhibited significant variations among the groups, with the highest abundance observed in group

H and the lowest in group CK. The relative abundance of ASVs corresponding to Gp6 was highest in group H, with only ASV_264 showing a significant difference. Furthermore, the abundance of ASV_64 in group H of *Aridibacter* was significantly higher than that in group CK. The analysis of the top 30 niche breadth type distribution of ASVs revealed a predominant presence of specialist species, followed by opportunist species, with no generalist species were observed. Interestingly, the relative abundance of ASV_15_*Arthrobacter* was high in all groups, with group H exhibiting a significantly higher relative abundance compared to group CK. However, this species was not observed in the inter-group distribution at the top 15 genus level (Fig. 6C and D).

## Effects of liquid fertilizer application on soil bacterial community function and plant growth

To investigate functional differences associated with taxonomic variations among groups treated with different concentrations of liquid fertilizer, predictive profiling was performed by Tax4Fun2. Rhizosphere bacterial function was predicted using Tax4Fun2 to explore the potential microbial functional differences between CK, L, and H groups during the different concentrations of liquid fertilizer period. Predicted functional pathways at level 2 according to KEGG reference for molecular functions of genes, including carbohydrate metabolism, biosynthesis of other secondary metabolites, immune system, replication and repair, folding, sorting and degradation, transcription, and immune diseases were found to be more abundant in liquid fertilizer treatment groups (L and H); in the CK group, rhizosphere predicted genes associated with xenobiotics biodegradation and metabolism, energy metabolism, infectious diseases: viral, endocrine, and metabolic diseases, substance dependence, and nervous system pathways were more abundant (Fig. 7A; Table S5).

The morphological parameters of bok choy after 32 days of planting under control and different concentrations of liquid fertilizer treatment were presented in Fig. 7B; Table

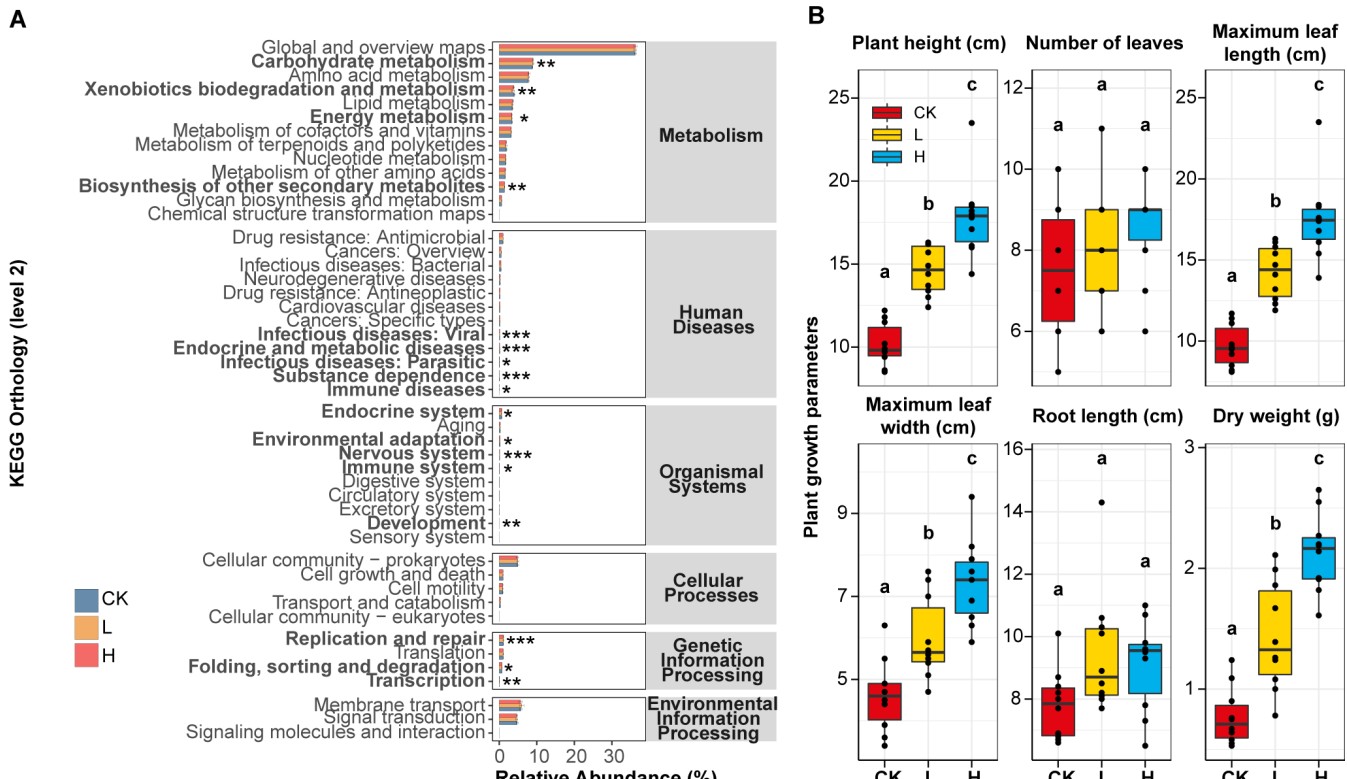

**FIG 7** Functional prediction of rhizosphere microbiota and bok choy growth parameters. (A) KEGG Pathway level 2 and level 3 function classification. (B) Indicators of plant, leaf, and root of bok choy. CK, unfertilized; L, low concentration; H, high concentration.

S6. The investigation of morphological traits indicated that high concentration of liquid fertilizer had significant effects on the plant growth. Compared with the CK group, the application of high-concentration liquid fertilizer (L group) significantly increased the plant height (average value 10.150 cm vs 17.820 cm), maximum leaf length (average value 9.710 cm vs 17.490 cm), maximum leaf width (average value 4.610 cm vs 7.350 cm), and dry weight (average value 0.771 g vs 2.126 g) of bok choy. Besides, even the low concentration liquid fertilizer treatment could significantly increase the above morphological indicators (Fig. 7B; Table S6).

## DISCUSSION

### Apple waste liquid fertilizer application significantly promoted the growth of plants

Among plant health parameters, plant height was considered a primary indicator of plant growth rate (48). Our study found that liquid fertilizer treatment could significantly increase the plant height of bok choy, and this effect was concentration-dependent. Our study also confirmed that the dry weight of bok choy increased significantly after applying liquid fertilizer, indicating that secondary metabolites in liquid fertilizer and soil microorganisms might work together to improve the biomass of bok choy (49, 50).

### Effects of taxa with different niche breadth on soil properties and community structure

In agro-ecosystems, microbial communities can provide a wide range of ecological functions and enhance soil multifunctionality (51). During plant growth, roots interacted with rhizosphere microorganisms, and the species' niche breadth reflected its utilization of environmental resources (52, 53). The findings of this study revealed significant associations between different taxa of niche breadth and distinct soil properties. It revealed significant correlations between niche breadth subgroups and soil AN as well as AP. Specifically, the specialist ASVs exhibited a stronger correlation with AP, while the generalist ASVs showed a stronger correlation with AN (Fig. 6). This difference might arise from variations in multifunctionality among sub-groups, with soil specialist ASVs involved in the mineralization of organic compounds, while generalist ASVs contributed to the release of fixed nutrients (54, 55). The macroecological pattern could be understood as the fluctuating of environmental resources, rather than a consequence of species competition (56). In our study, AN and AP were significantly positively correlated with distance based on Bray–Curtis dissimilarity, indicating that higher P and N content might promote the increase of abundance and, thus, affect the distribution of species.

Sphingomonas (ASV_22, ASV_37, ASV_14, and ASV_20), Gp6 (ASV_168, ASV_241, and ASV_264), and Arthrobacter (ASV_15) had high relative abundances across all samples, with ASV_37, ASV_14, ASV_264, ASV_15 showing a significant increase in group H (Fig. 6C and D). Sphingomonas, a class of probiotics, was recognized for their significant role in environmental remediation and promotion of plant growth (57). Moreover, Sphingomonas and Arthrobacter could be effectively recruited in an enrichment pattern to improve the resistance to disease (58). The abundance of Gp6, a member of Acidobacteria, exhibited a significant positive correlation with soil organic matter content, suggesting its involvement in the decomposition process of soil organic matter (59). In conjunction with our research, it was observed that liquid fertilizer, particularly in the high-concentration group, had the potential to alter the recruitment pattern of rhizosphere bacterial communities by modifying soil properties and enhancing the presence of beneficial or antagonistic bacteria. Consequently, this could influence the release of soil AN and AP, with a more pronounced impact attributed to the specialist ASVs.

## Liquid fertilizer affected the structure of bacterial communities

The presence of organic acids and soluble solids in liquid fertilizer could have specific impacts on the physical, chemical, and biological properties of soil (60). Our study found that different concentrations of apple waste liquid fertilizer significantly influence the structure and composition of rhizosphere bacterial communities. After 32 days of applying a high concentration liquid fertilizer, there was a significant increase observed in both the abundance and phylogenetic diversity of the root bacterial community (Fig. 2A). This enhancement can be attributed to the synergistic effects of beneficial microorganisms and organic acid metabolites present in the liquid fertilizer (60). The beta diversity Bray-Curtis PCoA showed evidence of change based upon both concentration and time. The results demonstrated a clear distinction in bacterial communities between the control group (CK) and the soil treated with liquid fertilizer after 32 days (Fig. 2B). The liquid fertilizer was reported to have dual functions as both a chemical regulator and a source of beneficial microorganisms and could also persist in the soil for an extended period, enhancing plant resistance during growth (61). The findings of several studies suggest that bacterial communities might exhibit limited sensitivity to soil changes resulting from agricultural cultivation (62). However, our study revealed a dynamic pattern in this regard. Specifically, we observed a significant decrease in Bray-Curtis dissimilarity on the 16th day after planting, followed by an increase back to the initial level (Fig. 2C). Interestingly, our study also revealed that a higher concentration of liquid fertilizer exhibits a significant reduction in Bray-Curtis dissimilarity, with this decline becoming more pronounced over time. Soil chemical properties and root microenvironment were crucial factors influencing the recruitment of microbiota by roots (63). The study revealed that the diversity of the root bacterial community increased, while Bray-Curtis dissimilarity decreased, indicating that altering the soil microenvironment through liquid fertilizer application might modify the process of bacterial recruitment in rhizosphere soil, leading to an earlier establishment of a diverse and stable bacterial community.

## Liquid fertilizer influenced the bacterial community by altering their niche breadth distribution

Land-use remains a consistent driver of soil community composition (64). Our results demonstrated that the application of liquid fertilizer did not significantly disrupt the process of community assembly, and deterministic processes governed the assembly regardless of the utilization of fertilizer during the planting (Fig. 3A). Soil pH was the most important edaphic variable mediating the changes in the bacterial community assembly processes, decreasing soil pH led to stochastic assembly of bacterial community (65). We also observed that after 32 days of high concentration liquid fertilizer application, the community transitioned into stochastic processes (with $\beta$NTI median <2).

Conversely, the application of apple waste liquid fertilizer had a greater impact on the niche breadth of bacterial community. Previous research showed that generalist species had more important roles in maintaining community stability when exposed to environmental disturbances (66). Our research also showed that an increased concentration of liquid fertilizer resulted in an expanded niche breadth, indicating a progression toward more generalist species with versatile functional capabilities. The process of niche construction involved the gradual development of colonial niches with time, which, in turn, gave rise to new potential sub-niches that impact species selection within various microbial communities (67). In this study, even short planting time (32 days) after application of liquid fertilizer significantly affected the niche breadth distribution of rhizosphere bacterial communities. The distribution of bacterial niche breadth in soil treated with a high concentration liquid fertilizer exhibited greater breadth, and this distribution significantly expanded as the duration of planting increased.

Generalization and specialization were evolutionarily stable with generalists supporting larger populations and specialists playing important roles within communities (68). In our research, specialists played important roles in community structure with

the highest proportion of relative abundance. Previous studies had suggested that plant roots could regulate the recruitment of beneficial microorganisms and facilitate host or its offspring survival (69, 70). The relative abundance of the antagonistic probiotics *Sphingomonas* and Gp6 was found to be significantly higher among the three niche breadth types (Fig. 4). In particular, the abundance of *Sphingomonas* in the specialist group significantly increased with the concentration and timing of liquid fertilizer application, indicating that apple waste liquid fertilizer could influence the distribution and quantity of bacterial species across different categories of bacterial niche breadth by modulating the soil microenvironment.

## The concentration primarily influenced the specialists of the bacterial community co-occurrence network

Niche breadth had a clear and profound effect on the relationship between environmental heterogeneity and species co-occurrence (55). In this study, despite specialist abundance constituting the majority of rhizosphere bacteria, the co-occurrence pattern of indicators exhibited significant alterations following the application of liquid fertilizer. Subsequent to the application of liquid fertilizer compared to the CK group, the bacterial network exhibited a decrease in the number of nodes and edges, as well as an increase in modularity (Fig. 5).

A previous study found that communities with more specialists had fewer linkages in co-occurrence network than those with more generalists (71). In this study, it was observed that although there was no significant difference in proportion, the liquid fertilizer application groups exhibited a higher prevalence of specialist subgroups with centrality and their relative abundance was higher. Species with centrality were generally considered to be the core taxa or the hub taxa of clusters in a network (72, 73). Our findings indicated a significant increase in taxa with centrality following the application of liquid fertilizer, which exhibited a positive correlation with concentration. And members of *Sphingomonas* were identified as both hubs and connectors in liquid fertilizer application groups, suggesting their role as keystone groups. The properties of soil and improvement strategies would impact microbial network structure and modularity (74, 75). Our network analyses revealed that co-occurrence patterns of bacterial ASVs changed across concentration gradients of liquid fertilizer and were likely to be mediated by soil environmental changes (76).

## Liquid fertilizer application significantly affected the function of soil bacteria community

Functional prediction was performed to better understand the implications of the differences identified in bacterial community composition and interaction under liquid fertilizer treatment. The biosynthetic role of glycolysis and respiration was particularly important in actively growing autotrophic tissues (77). In our study, carbohydrate metabolism and biosynthesis of other secondary metabolites pathway gene significantly increased in liquid fertilizer treatment might indicate that the carbon source provided by liquid fertilizer could be converted into various secondary metabolites by soil microorganisms. Specific agricultural management practices can either promote or interfere with soil's ability to regulate pathogens (78). The KEGG functional prediction results revealed that high concentration of liquid fertilizer significantly reduced the abundance of human disease functional gene pathways in the soil, including infectious diseases: viral, endocrine, and metabolic diseases, and substance dependence.

Different functional pathways could lead to different physiological consequences. In the most abundant functional pathways involved in metabolism (L1), both the L and H groups showed enhanced activity in energy metabolism as well as xenobiotics biodegradation and metabolism. It is inferred that the enzymes in liquid fertilizer could enhance the activity of bacterial biodegradation to remove xenobiotics, which might need to mobilize metabolism routes linked to energy (79). In contrast to the L and H group, CK had higher abundances of carbohydrate metabolism and biosynthesis of

other secondary metabolites. This might be attributed to the enzyme system in the liquid fertilizer, such as protease and cellulase, which accelerated the decomposition of organic matter (80). In contrast, the soil of the CK group exhibited greater accumulation of macromolecular carbohydrates, necessitating a more active hydrolase system. This was reflected in the increased enzyme activity and gene abundance associated with carbohydrate metabolism.

## Conclusions

This study revealed the succession of bacterial communities following different concentrations of apple waste liquid fertilizer treatment during planting, while also highlighting the impact of bacterial communities on soil properties. The Chao1 index, phylogenetic diversity index, beta diversity distance, and Bray-Curtis dissimilarity of bacteria exhibited significant differences at high concentrations, especially during the later stages of planting. The concentration of liquid fertilizer was also found to have a significant impact on the distribution of niche breadth among rhizosphere bacteria. The rhizosphere bacterial communities exhibited reduced network complexity, including a decrease in the number of nodes, edges, and overall network size following the application of liquid fertilizer. The number of central specialist indicators of high-abundance ASVs was increased following the use of liquid fertilizer. In particular, *Sphingomonas* were identified as hubs and connectors within the key indicators in the groups where liquid fertilizer was applied. The study also found that specialist ASVs were significantly correlated with AP, while generalist ASVs were significantly correlated with AN. Furthermore, liquid fertilizer treatment significantly affected the metabolism and genetic information processing of soil bacterial community and significantly promoted the growth of plants.

While this study contributes to understanding the influences of apple waste liquid fertilizer on soil ecology and plant growth, it has certain limitations. First, the combined effects of soil biota on soil ecology resulted in significant alterations, with this study primarily focusing on the impacts of bacterial communities. Second, this research mainly assessed the short-term impacts of the liquid phase of apple wastes on soil and plants within a low-concentration range. Despite these limitations, these findings had expanded our understanding of the interaction between bacterial communities and soil properties during planting with varying concentrations of apple wastes liquid fertilizer, thereby providing new insights into bacteria regulation of rhizosphere microhabitats.

## ACKNOWLEDGMENTS

This study was supported by Science and Technology Project of Shaanxi Academy of Science (2024k-11).

## AUTHOR AFFILIATIONS

[1]Shaanxi Institute of Microbiology, Xi'an, China
[2]Institute of Medical Research, Northwestern Polytechnical University, Xi'an, Shaanxi, China
[3]Department of Clinical Laboratory, Northwest Women's and Children's Hospital, Xi'an, China

## AUTHOR ORCIDs

Fengan Jia  https://orcid.org/0009-0006-9179-6025
Fan Chang  http://orcid.org/0000-0002-6422-1852

## FUNDING

| Funder | Grant(s) | Author(s) |
|---|---|---|
| Science and technology project of Shaanxi Academy of Science | | Fan Chang |

## AUTHOR CONTRIBUTIONS

Fengan Jia, conceptualization, Writing – original draft | Fan Chang, Writing – review and editing | Qingan Jia, Data curation | Yan Li, visualization | Lisha Zhen, investigation | Rui Lv, validation | Yun Xie, Writing – original draft, Writing – review and editing

## ADDITIONAL FILES

The following material is available online.

### Supplemental Material

**Fig. S1 (Spectrum02068-24-s0001.tif).** Graphic abstract.
**Supplemental tables (Spectrum02068-24-s0002.docx).** Tables S1 to S7.

### Open Peer Review

**PEER REVIEW HISTORY (review-history.pdf).** An accounting of the reviewer comments and feedback.

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
