## [Reviewer comments · Microbiology Spectrum]

Microbiology Spectrum

Responses of rhizosphere bacterial communities with different niche breadths to liquid fertilizer produced from Fuji apple wastes during planting process

Fengan Jia, Fan Chang, Qingan Jia, Yan Li, Lisha Zhen, Rui Lv, and Yun Xie

Corresponding Author(s): Fan Chang, Shaanxi Institute of Microbiology

Review Timeline:

Submission Date:	August 15, 2024
Editorial Decision:	November 5, 2024
Revision Received:	January 5, 2025
Editorial Decision:	February 28, 2025
Revision Received:	April 13, 2025
Editorial Decision:	April 21, 2025
Revision Received:	April 24, 2025
Accepted:	May 7, 2025

Editor: Philips Akinwale

Reviewer(s): The reviewers have opted to remain anonymous.

Transaction Report:

DOI: <https://doi.org/10.1128/spectrum.02068-24>

Re: Spectrum02068-24 (Responses of rhizosphere bacterial communities with different niche breadths to liquid fertilizer produced from Fuji apple wastes during planting process)

Dear Mr. Fan Chang:

Thank you for the privilege of reviewing your work. Below you will find my comments, instructions from the Spectrum editorial office, and the reviewer comments.

Revision Guidelines

Sincerely,
Philips Akinwale
Editor
Microbiology Spectrum

Reviewer #2 (Public repository details (Required)):

Sequence data need to be deposited in a publicly available database

Reviewer #2 (Comments for the Author):

Thank you for the opportunity to review the manuscript titled "Responses of rhizosphere bacterial communities with different niche breadths to liquid fertilizer produced from Fuji apple wastes during planting process." This study sought to examine the effects of liquid fertilizer from apple waste on soil rhizosphere microbial community structure and soil properties. The established replicated experimental plots with bok choy that was amended with low and high concentrations of liquid fertilizer. Significant differences were observed in microbial diversity, abundance, and niche breadth across treatments and time but not community assembly. Key bacterial indicators like *Sphingomonas* were identified. Specialist, generalist, and opportunist species were identified and compared across treatments and time. Overall, this was a very interesting study with important implications for agriculture. Appropriate and interesting statistical approaches were employed to analyze the data. More details need to be provided regarding the methodology and experimental design. I think the discussion also needs to be expanded a bit with additional focus on the ecology and function of individual taxa. There was also not much information on changes in soil properties over time and/or with the addition of fertilizer which seemed to be a key investigation of the study. The writing was a bit difficult to follow in the introduction and discussion sections and could benefit from additional proofreading and editing. Additional, and specific, comments and recommendations are included in the attached annotated pdf.

Reviewer #3 (Comments for the Author):

This study investigated the effects of apple waste liquid fertilizer at different concentrations on soil rhizosphere bacterial community structure and assembly processes, demonstrating that high-concentration treatment significantly altered bacterial community assembly from stochastic to deterministic processes, increased specialist abundance, and formed tighter microbial interaction networks, providing new insights for agricultural waste utilization and soil microbial community regulation.

Areas Requiring Improvement:

1. Experimental Section:

- Include detailed nutritional composition analysis of liquid fertilizer
- Add plant growth parameter measurements
- Consider extending observation period (i.e. 1 year) for long-term effects

2. It lacks functional analyses such as:

- Metabolic pathway prediction
- Functional gene abundance
- Enzyme activities
- Nutrient cycling processes
- Specific roles of key bacterial groups in soil improvement
- Plant growth-promoting traits of identified specialists

3. While the current study focuses mainly on microbial community structural changes, it would benefit from the following key additions to make it more comprehensive: First, include basic fertilizer properties (NPK content, pH), key soil enzyme activities (such as urease and phosphatase), and basic plant growth parameters (plant height, biomass); Second, for the microbial aspects, beyond the existing community structure analysis, incorporate functional gene sequencing or prediction analysis, focusing on nutrient cycling (nitrogen and phosphorus cycles) and plant growth promotion (such as IAA synthesis, phosphate solubilization) pathways. These additions would better explain the fertilizer-microbe-plant interaction mechanisms and enhance the study's practical application value.

4. Writing Style:

- Simplify complex sentences
- Improve clarity and conciseness

For example : Lines 27-29: "The rhizosphere bacterial communities treated with a high concentration of liquid fertilizer exhibited increased diversity and broader niche breadth distribution during the late planting period." The reviewer suggested it to two sentences : "High concentration liquid fertilizer treatment altered rhizosphere bacterial communities. These communities showed increased diversity and broader niche breadth distribution in the late planting period."

[revised manuscript text omitted]

**Reference**

[1] Philippot, L., Chenu, C., Kappler, A., Rillig, M.C., Fierer, N., 2024.
The interplay between microbial communities and soil properties. *Nat Rev*
*Microbiol*, 2024, **22**, 226–239.

[2] Peng, Z., Qian, X., Liu, Y., Li, X., et al., 2024. Land conversion to
agriculture induces taxonomic homogenization of soil microbial communities
globally. *Nat Commun*, 2024, **15**, 3624.

[3] Jing, J., Cong, W.-F., Bezemer, T.M., 2022. Legacies at work: plant–
soil–microbiome interactions underpinning agricultural sustainability. *Trends*
*Plant Sci*, 2022, **27**, 781–792.

[4] Wu, H., Cui, H., Fu, C., Li, R., et al., 2024. Unveiling the crucial
role of soil microorganisms in carbon cycling: A review. *Science of The Total*
*Environment*, 2024, **909**, 168627.

[5] Hartmann, M., Six, J., 2022. Soil structure and microbiome
functions in agroecosystems. *Nat Rev Earth Env*, 2022, **4**, 4–18.

[6] El-Saadony, M.T., Saad, A.M., Soliman, S.M., Salem, H.M., et al.,

2022. Plant growth-promoting microorganisms as biocontrol agents of plant
diseases: Mechanisms, challenges and future perspectives. *Front Plant Sci*, 2022,
**13**, 923880.

[7] Shi, G., Sun, H., Calderón-Urrea, A., Li, M., et al., 2021. Bacterial
communities as indicators of soil health under a continuous cropping system.
*Land Degrad Dev*, 2021, **32**, 2393–2408.

[8] Fierer, N., Wood, S.A., Bueno de Mesquita, C.P., 2021. How
microbes can, and cannot, be used to assess soil health. *Soil Biology and*
*Biochemistry*, 2021, **153**, 108111.

[9] Thepbandit, W., Athinuwat, D., 2024. Rhizosphere Microorganisms
Supply Availability of Soil Nutrients and Induce Plant Defense. *Microorganisms*,
2024, **12**, 558.

[10] Schmidt, J.E., Vannette, R.L., Igwe, A., Blundell, R., et al., 2019.
Effects of Agricultural Management on Rhizosphere Microbial Structure and
Function in Processing Tomato Plants. *Appl Environ Microb*, 2019, **85**,
e01064-19.

[11] Wang, T., Yang, K., Ma, Q., Jiang, X., et al., 2022. Rhizosphere
Microbial Community Diversity and Function Analysis of Cut Chrysanthemum
During Continuous Monocropping. *Front Microbiol*, 2022, **13**, 801546.

[12] Toor, G.S., Yang, Y.-Y., Das, S., Dorsey, S., Felton, G., 2021.
*Advances in Agronomy*, vol. 168, Elsevier, pp. 157–201.

[13] De Corato, U., 2020. Agricultural waste recycling in horticultural
intensive farming systems by on-farm composting and compost-based tea
application improves soil quality and plant health: A review under the
perspective of a circular economy. *Science of The Total Environment*, 2020, **738**,
139840.

[14] Liu, J., Shu, A., Song, W., Shi, W., et al., 2021. Long-term organic
fertilizer substitution increases rice yield by improving soil properties and
regulating soil bacteria. *Geoderma*, 2021, **404**, 115287.

[15] Garbowski, T., Bar-Michalczyk, D., Charazińska, S.,
Grabowska-Polanowska, B., et al., 2023. An overview of natural soil
amendments in agriculture. *Soil and Tillage Research*, 2023, **225**, 105462.

[16] Elnahal, A.S.M., El-Saadony, M.T., Saad, A.M., Desoky, E.-S.M., et
al., 2022. The use of microbial inoculants for biological control, plant growth
promotion, and sustainable agriculture: A review. *Eur J Plant Pathol*, 2022, **162**,
759–792.

[17] Mengqi, Z., Shi, A., Ajmal, M., Ye, L., Awais, M., 2023.
Comprehensive review on agricultural waste utilization and high-temperature
fermentation and composting. *Biomass Convers Bior*, 2023, **13**, 5445–5468.

[18] Martín-Lammerding, D., Gabriel, J.L., Zambrana, E.,
Santín-Montanyá, I., Tenorio, J.L., 2021. Organic Amendment vs. Mineral
Fertilization under Minimum Tillage: Changes in Soil Nutrients, Soil Organic

- Matter, Biological Properties and Yield after 10 Years. *Agriculture*, 2021, **11**,
700.
- [19] Dahunsi, S.O., Oranusi, S., Efeovbokhan, V.E., Adesulu-Dahunsi,
629 A.T., Ogunwole, J.O., 2021. Crop performance and soil fertility improvement
using organic fertilizer produced from valorization of Carica papaya fruit peel.
*Sci Rep-uk*, 2021, **11**, 4696.
- [20] Bello, S.K., Alayafi, A.H., AL-Solaimani, S.G., Abo-Elyousr,
633 K.A.M., 2021. Mitigating Soil Salinity Stress with Gypsum and Bio-Organic
Amendments: A Review. *Agronomy*, 2021, **11**, 1735.
- [21] Anyaoha, K.E., Sakrabani, R., Patchigolla, K., Mouazen, A.M., 2018.
Critical evaluation of oil palm fresh fruit bunch solid wastes as soil amendments:
Prospects and challenges. *Resour Conserv Recy*, 2018, **136**, 399–409.
- [22] Ramamoorthy, K., Dhanraj, R., Vijayakumar, N., Ma, Y., et al., 2024.
Vegetable and fruit wastes: Valuable source for organic fertilizer for effective
growth of short-term crops: *Solanum lycopersicum* and *Capsicum annum*.
*Environ Res*, 2024, **251**, 118727.
- [23] Puspitawati, M.D., Sumiasih, I.H., 2021. Organic Fertilizer from
Starfruit Waste Sustainable Agriculture Solution. *IOP Conf. Ser.: Earth Environ.*
*Sci.*, 2021, **709**, 012069.
- [24] Macias-Benitez, S., Garcia-Martinez, A.M., Caballero Jimenez, P.,
Gonzalez, J.M., et al., 2020. Rhizospheric Organic Acids as Biostimulants:
Monitoring Feedbacks on Soil Microorganisms and Biochemical Properties.
*Front Plant Sci*, 2020, **11**, 633.
- [25] Becker, W., Helsing, E. (Eds.), 1991. Food and health data: their use
in nutrition policy-making., World Health Organization, Regional Office for
Europe, Copenhagen.
- [26] Logue, J.B., Stedmon, C.A., Kellerman, A.M., Nielsen, N.J., et al.,
2016. Experimental insights into the importance of aquatic bacterial community
composition to the degradation of dissolved organic matter. *ISME J*, 2016, **10**,
533–545.
- [27] Edgar, R.C., 2010. Search and clustering orders of magnitude faster
than BLAST. *Bioinformatics*, 2010, **26**, 2460–2461.
- [28] Edgar, R.C., 2018. Updating the 97% identity threshold for 16S
ribosomal RNA OTUs. *Bioinformatics*, 2018, **34**, 2371–2375.
- [29] Knight, R., Vrbanac, A., Taylor, B.C., Aksenov, A., et al., 2018. Best
practices for analysing microbiomes. *Nature Reviews Microbiology*, 2018, **16**,
410–422.
- [30] Bacci, G., Bani, A., Bazzicalupo, M., Ceccherini, M.T., et al., 2015.
Evaluation of the Performances of Ribosomal Database Project (RDP) Classifier
for Taxonomic Assignment of 16S rRNA Metabarcoding Sequences Generated
from Illumina-Solexa NGS. *J. Genomics*, 2015, **3**, 36–39.
- [31] Qu, B., Liu, Y., Sun, X., Li, S., et al., 2019. Effect of various

- mulches on soil physico—Chemical properties and tree growth (*Sophora*
*japonica*) in urban tree pits. PLoS ONE, 2019, **14**, e0210777.
- [32] Bradstreet, R.B., 1954. Kjeldahl Method for Organic Nitrogen.
Analytical Chemistry, 1954, **26**, 185–187.
- [33] Khan, S.A., Mulvaney, R.L., Mulvaney, C.S., 1997. Accelerated
Diffusion Methods for Inorganic-Nitrogen Analysis of Soil Extracts and Water.
Soil Science Society of America Journal, 1997, **61**, 936–942.
- [34] Murphy, J., Riley, J.P., 1962. A modified single solution method for
the determination of phosphate in natural waters. Analytica Chimica Acta, 1962,
**27**, 31–36.
- [35] Wolf, A.M., Baker, D.E., 1985. Comparisons of soil test phosphorus
by Olsen, Bray P1, Mehlich I and Mehlich III methods. Communications in Soil
Science and Plant Analysis, 1985, **16**, 467–484.
- [36] Attoe, O.J., Truog, E., 1947. Rapid Photometric Determination of
Exchangeable Potassium and Sodium. Soil Science Society of America Journal,
1947, **11**, 221–226.
- [37] Fan, K., Weisenhorn, P., Gilbert, J.A., Shi, Y., et al., 2018. Soil pH
correlates with the co-occurrence and assemblage process of diazotrophic
communities in rhizosphere and bulk soils of wheat fields. Soil Biology and
Biochemistry, 2018, **121**, 185–192.
- [38] Chen, T., Liu, Y., Huang, L., 2022. ImageGP: An easy-to-use data
visualization web server for scientific researchers. IMeta, 2022, **1**.
- [39] Oksanen, J., Blanchet, F.G., Friendly, M., Kindt, R., et al., 2019.
vegan: Community Ecology Package.
- [40] Kembel, S.W., Cowan, P.D., Helmus, M.R., Cornwell, W.K., et al.,
2010. Picante: R tools for integrating phylogenies and ecology. Bioinformatics,
2010, **26**, 1463–1464.
- [41] Stegen, J.C., Lin, X., Konopka, A.E., Fredrickson, J.K., 2012.
Stochastic and deterministic assembly processes in subsurface microbial
communities. ISME J, 2012, **6**, 1653–1664.
- [42] Levins, R., 1968. Evolution in changing environments: some
theoretical explorations. Monographs in Population Biology, 1968.
- [43] Wu, W., Lu, H.-P., Sastri, A., Yeh, Y.-C., et al., 2018. Contrasting the
relative importance of species sorting and dispersal limitation in shaping marine
bacterial versus protist communities. The ISME Journal, 2018, **12**, 485–494.
- [44] Cáceres, M.D., Legendre, P., 2009. Associations between species
and groups of sites: indices and statistical inference. Ecology, 2009, **90**, 3566–
3574.
- [45] Bastian, M., Heymann, S., Jacomy, M., 2009. Gephi: An Open
Source Software for Exploring and Manipulating Networks. ICWSM, 2009, **3**,
361–362.
- [46] Diniz-Filho, J.A.F., Soares, T.N., Lima, J.S., Dobrovolski, R., et al.,

2013. Mantel test in population genetics. *Genet Mol Biol.*, 2013, **36**, 475–485.

[47] Gao, Y., Zhang, Y., Cheng, X., Zheng, Z., et al., 2022. Agricultural
Jiaosu: An Eco-Friendly and Cost-Effective Control Strategy for Suppressing
*Fusarium Root Rot Disease in Astragalus membranaceus*. *Front Microbiol*, 2022,
**13**, 823704.

[48] Cheng, X., Gao, Y., Wang, Z., Cai, Y., Wang, X., 2023. Agricultural
Jiaosu Enhances the Stress Resistance of Pak Choi (*Brassica rapa L. subsp.*
*chinensis*) by Recruiting Beneficial Rhizosphere Bacteria and Altering Metabolic
Pathways. *Agronomy*, 2023, **13**, 2310.

[49] Coller, E., Oliveira Longa, C.M., Morelli, R., Zanoni, S., et al., 2021.
Soil Communities: Who Responds and How Quickly to a Change in Agricultural
System? *Sustainability-basel*, 2021, **14**, 383.

[50] Beschoren da Costa, P., Benucci, G.M.N., Chou, M.-Y., Van Wyk, J.,
et al., 2022. Soil Origin and Plant Genotype Modulate Switchgrass Aboveground
Productivity and Root Microbiome Assembly. *Mbio*, 2022, **13**, e00079-22.

[51] Barnett, S.E., Youngblut, N.D., Buckley, D.H., 2020. Soil
characteristics and land-use drive bacterial community assembly patterns. *FEMS*
*Microbiology Ecology*, 2020, **96**, fiz194.

[52] Zhou, X., Khashi u Rahman, M., Liu, J., Wu, F., 2021. Soil
acidification mediates changes in soil bacterial community assembly processes
in response to agricultural intensification. *Environ Microbiol*, 2021, **23**, 4741–
4755.

[53] Xu, Q., Vandenkoornhuyse, P., Li, L., Guo, J., et al., 2022. Microbial
generalists and specialists differently contribute to the community diversity in
farmland soils. *Journal of Advanced Research*, 2022, **40**, 17–27.

[54] Baquero, F., Coque, T.M., Galán, J.C., Martinez, J.L., 2021. The
Origin of Niches and Species in the Bacterial World. *Front Microbiol*, 2021, **12**,
657986.

[55] Hernandez, D.J., Kieseewetter, K.N., Almeida, B.K., Revillini, D.,
Afkhami, M.E., 2023. Multidimensional specialization and generalization are
pervasive in soil prokaryotes. *Nat Ecol Evol*, 2023, **7**, 1408–1418.

[56] Gao, M., Xiong, C., Gao, C., Tsui, C.K.M., et al., 2021.
Disease-induced changes in plant microbiome assembly and functional
adaptation. *Microbiome*, 2021, **9**, 187.

[57] Compant, S., Clément, C., Sessitsch, A., 2010. Plant
growth-promoting bacteria in the rhizo- and endosphere of plants: Their role,
colonization, mechanisms involved and prospects for utilization. *Soil Biology*
*and Biochemistry*, 2010, **42**, 669–678.

[58] Bar-Massada, A., 2015. Complex relationships between species
niches and environmental heterogeneity affect species co-occurrence patterns in
modelled and real communities. *P Roy Soc B-biol Sci*, 2015, **282**, 20150927.

[59] Liu, C., Li, X., Mansoldo, F.R.P., An, J., et al., 2022. Microbial

habitat specificity largely affects microbial co-occurrence patterns and functional
profiles in wetland soils. *Geoderma*, 2022, **418**, 115866.

[60] Cardinale, M., Ratering, S., Sadeghi, A., Pokhrel, S., et al., 2020.
The Response of the Soil Microbiota to Long-Term Mineral and Organic
Nitrogen Fertilization is Stronger in the Bulk Soil than in the Rhizosphere.
*Genes-basel*, 2020, **11**, 456.

[61] Ye, Z., Wang, J., Li, J., Liu, G., et al., 2022. Different roles of core
and noncore bacterial taxa in maintaining soil multinutrient cycling and
microbial network stability in arid fertigation agroecosystems. *J Appl Ecol*, 2022,
**59**, 2154–2165.

[62] Ishimoto, C.K., Aono, A.H., Nagai, J.S., Sousa, H., et al., 2021.
Microbial co-occurrence network and its key microorganisms in soil with
permanent application of composted tannery sludge. *Science of The Total
Environment*, 2021, **789**, 147945.

[63] Guo, Y., Song, B., Li, A., Wu, Q., et al., 2022. Higher pH is
associated with enhanced co-occurrence network complexity, stability and
nutrient cycling functions in the rice rhizosphere microbiome. *Environmental
Microbiology*, 2022, **24**, 6200–6219.

[64] Lin, Q., Li, L., Adams, J.M., Heděnc, P., et al., 2021. Nutrient
resource availability mediates niche differentiation and temporal co-occurrence
of soil bacterial communities. *Appl Soil Ecol*, 2021, **163**, 103965.

[65] Jiao, S., Yang, Y., Xu, Y., Zhang, J., Lu, Y., 2020. Balance between
community assembly processes mediates species coexistence in agricultural soil
microbiomes across eastern China. *ISME J*, 2020, **14**, 202–216.

[66] Favela, A., O. Bohn, M., D. Kent, A., 2021. Maize germplasm
chronosequence shows crop breeding history impacts recruitment of the
rhizosphere microbiome. *ISME J*, 2021, **15**, 2454–2464.

[67] Sireci, M., Muñoz, M.A., Grilli, J., 2023. Environmental
fluctuations explain the universal decay of species-abundance correlations with
phylogenetic distance. *Proc. Natl. Acad. Sci. U.S.A.*, 2023, **120**, e2217144120.

[68] Asaf, S., Numan, M., Khan, A.L., Al-Harrasi, A., 2020.
*Sphingomonas* : from diversity and genomics to functional role in environmental
remediation and plant growth. *Crit Rev Biotechnol*, 2020, **40**, 138–152.

[69] Liu, S., Liu, R., Zhang, S., Shen, Q., et al., 2024. The Contributions
of Sub-Communities to the Assembly Process and Ecological Mechanisms of
Bacterial Communities along the Cotton Soil–Root Continuum Niche Gradient.
*Microorganisms*, 2024, **12**, 869.

[70] Ge, Z., Li, S., Bol, R., Zhu, P., et al., 2021. Differential long-term
fertilization alters residue-derived labile organic carbon fractions and microbial
community during straw residue decomposition. *Soil and Tillage Research*, 2021,
**213**, 105120.

Table S1 Alpha indices of bacterial community in the study. All values were an average from all replicates \pm standard deviations (SD).

Groups	Days	Chao1	Shannon	Faith PD	Good coverage
CK	2	6271.113 \pm 1715.567	7.285 \pm 0.903	357.624 \pm 82.081	0.983 \pm 0.004
	16	10248.522 \pm 1720.338	8.067 \pm 0.067	459.387 \pm 17.146	0.961 \pm 0.015
	32	6928.006 \pm 3074.215	7.568 \pm 0.574	382.932 \pm 83.897	0.979 \pm 0.013
L	2	8061.217 \pm 3084.731	7.007 \pm 1.095	410.794 \pm 78.804	0.971 \pm 0.014
	16	8400.841 \pm 2929.293	7.724 \pm 0.354	418.889 \pm 65.369	0.97 \pm 0.018
	32	9823.701 \pm 1187.183	7.56 \pm 0.474	443.571 \pm 19.437	0.961 \pm 0.012
H	2	9653.49 \pm 3082.299	7.764 \pm 0.531	439.574 \pm 65.144	0.962 \pm 0.018
	16	10278.691 \pm 1634.74	7.872 \pm 0.334	470.05 \pm 24.116	0.963 \pm 0.012
	32	11244.7 \pm 404.442	7.88 \pm 0.211	461.99 \pm 10.038	0.952 \pm 0.004

Table S2 Bray–Curtis dissimilarity, β NTI and niche breadths of bacterial community in the study. All values were an average from all replicates \pm standard deviations.

Groups	Days	Bray–Curtis dissimilarity	β NTI	Niche Breadth
CK	2	0.702 \pm 0.161	3.029 \pm 1.490	1.949 \pm 0.969
	16	0.442 \pm 0.076	4.948 \pm 1.906	3.104 \pm 1.287
	32	0.662 \pm 0.117	3.045 \pm 1.180	2.052 \pm 1.006
L	2	0.659 \pm 0.132	2.311 \pm 1.495	2.297 \pm 1.031
	16	0.597 \pm 0.101	3.167 \pm 1.709	2.305 \pm 1.065
	32	0.49 \pm 0.07	3.389 \pm 1.299	2.802 \pm 1.290
H	2	0.548 \pm 0.207	4.964 \pm 3.022	2.645 \pm 1.184

16	0.506±0.069	3.806±2.280	2.852±1.227
32	0.367±0.058	1.571±1.890	3.248±1.415

Table S3 Relative abundance of different groups and different niche breadths in the top 10 phyla and the top 15 genera.

Niche breadth	Species taxonomic level	CK (%)			L (%)			H (%)		
		Day2	Day16	Day32	Day2	Day16	Day32	Day2	Day16	Day32
	Phylum									
SPECIALIST	Alphaproteobacteria	11.21	13.05	12.21	18.52	14.21	19.18	15.35	13.43	18.63
	Gammaproteobacteria	9.33	6.85	8.07	15.05	9.35	7.77	8.24	10.42	9.39
	Actinobacteria	7.83	5.35	7.55	7.12	7.29	9.44	5.22	5.05	6.18
	Acidobacteria	9.18	7.71	7.63	4.66	5.52	4.48	7.19	6.57	4.74
	Bacteroidetes	4.24	4.27	6.17	4.23	5.4	5.46	5.3	3.63	5.2
	Planctomycetes	7.63	3.85	6.72	4.35	4.59	2.51	3.96	4.8	1.66
	Deltaproteobacteria	4.68	4.48	5.51	3.38	5.7	3.19	3.12	4.51	3.14
	Betaproteobacteria	5.26	2.75	4.2	2.77	3.65	3.7	3.96	2.97	4.73
	Firmicutes	2.82	1.39	2.22	5.72	4.04	4.1	2.35	3.15	2.31
	Verrucomicrobia	1.6	0.97	1.22	1	1.33	0.91	0.74	1.04	0.74
	Others	5.61	2.17	3.67	2.67	3.09	2.58	2.93	2.77	2.49
OPPORTUNIST	Acidobacteria	6	9	6.42	5.09	5.9	5.52	8.33	7.23	6.02
	Alphaproteobacteria	3.19	5.52	4.07	3.81	4.25	4.87	4.89	4.59	5.73
	Planctomycetes	4.9	4.57	4.65	3.9	4.42	3.05	3.76	5.23	2.33
	Actinobacteria	2.54	4.6	2.79	2.41	3.07	3.58	3.57	3.11	4.06
	Betaproteobacteria	1.97	3.31	2.42	2	2.39	2.72	2.95	2.68	3.18
	Gammaproteobacteria	1.71	2.33	1.74	1.57	1.79	1.55	1.91	1.99	1.87
	Deltaproteobacteria	1.12	2.15	1.63	1.27	1.65	1.53	1.84	1.94	1.96
	Firmicutes	1.22	1.62	1.37	1.37	1.63	1.65	1.55	1.92	1.41
	Bacteroidetes	0.8	1.48	1.08	1.04	1.26	1.56	1.16	1.26	1.95
	Verrucomicrobia	0.63	1.19	0.9	0.75	0.96	1.07	0.94	1.23	1.31
	Others	1.76	3.1	2.39	1.91	2.39	2.6	3.07	3.06	2.9
GENERALIST	Actinobacteria	0.75	1.48	0.73	0.78	0.93	1.17	1.22	1.04	1.38
	Acidobacteria	0.62	1.34	0.79	0.78	0.89	0.97	1.3	1.19	1.16
	Firmicutes	0.78	0.96	0.9	0.8	0.9	0.98	0.93	0.97	0.98
	Alphaproteobacteria	0.48	0.97	0.55	0.63	0.65	0.91	0.94	0.87	1.18
	Gammaproteobacteria	0.38	0.5	0.39	0.42	0.49	0.46	0.5	0.5	0.51
	Betaproteobacteria	0.29	0.55	0.37	0.38	0.43	0.52	0.52	0.49	0.56
	Bacteroidetes	0.34	0.5	0.37	0.38	0.4	0.41	0.48	0.42	0.5
	Deltaproteobacteria	0.23	0.46	0.24	0.24	0.32	0.37	0.39	0.47	0.46
	Planctomycetes	0.25	0.4	0.31	0.32	0.31	0.28	0.37	0.4	0.27
	Unassigned	0.14	0.28	0.16	0.17	0.2	0.21	0.26	0.24	0.25

	Others	0.45	0.83	0.57	0.5	0.6	0.69	0.73	0.79	0.82
	Genus (%)									
SPECIALIST	Sphingomonas	1.96	3.94	2.72	2.24	2.68	4.39	3.67	2.96	5.63
	Gp6	4.51	3.46	3.35	1.97	2.21	1.7	3.26	3.02	2
	Pseudoxanthomonas	0.3	0.25	0.38	6.42	1.31	1.74	1.87	1.51	1.7
	Bacillus	0.9	0.4	0.95	2.21	2.32	2.51	0.69	1.75	1.31
	Gemmata	1.9	1.06	1.77	0.92	1.02	0.67	1.06	1.15	0.35
	Lysobacter	0.67	0.92	0.49	0.74	0.91	1.19	1.26	1	2.17
	Pirellula	1.6	0.92	1.42	0.98	0.96	0.5	0.94	1.11	0.4
	Rhizobium	0.4	0.54	0.39	1.39	1.26	1.68	1.44	0.67	1.01
	Pseudomonas	0.78	0.51	0.52	1.31	0.98	1.15	1.09	0.77	0.98
	Ohtaekwangia	1.06	1.06	1.27	0.55	1.36	0.4	0.49	0.85	0.93
	Bradyrhizobium	1.07	0.89	0.96	1.4	0.69	0.45	0.59	0.81	0.52
	Povalibacter	1.66	1	1.67	0.48	0.88	0.22	0.44	0.55	0.34
	Aridibacter	0.37	0.76	0.63	0.49	0.5	0.48	0.81	0.69	0.7
	Devosia	0.2	0.5	0.41	0.71	0.77	0.94	0.42	0.59	0.67
	Ensifer	0.41	0.37	0.38	0.56	0.56	0.54	0.86	0.56	0.73
	Others	51.65	36.26	47.84	47.11	45.77	44.78	39.5	40.37	39.76
OPPORTUNIST	Gp6	3.93	5.54	3.93	3.12	3.55	2.96	5.07	4.22	3.27
	Pirellula	0.84	0.87	0.8	0.73	0.75	0.48	0.7	0.9	0.45
	Gemmata	0.88	0.78	0.9	0.61	0.77	0.57	0.59	0.9	0.34
	Gemmatimonas	0.32	0.8	0.52	0.44	0.51	0.8	0.75	0.68	0.77
	Gp4	0.32	0.72	0.52	0.42	0.52	0.57	0.86	0.66	0.63
	Neisseria	0.56	0.57	0.52	0.53	0.54	0.6	0.59	0.57	0.57
	Zavarzinella	0.54	0.41	0.47	0.38	0.44	0.26	0.4	0.54	0.19
	Gaiella	0.38	0.58	0.39	0.31	0.34	0.34	0.49	0.35	0.35
	Sphingomonas	0.19	0.46	0.28	0.22	0.37	0.39	0.32	0.33	0.46
	Bacillus	0.26	0.22	0.2	0.32	0.38	0.46	0.23	0.46	0.34
	Blastopirellula	0.41	0.45	0.37	0.34	0.32	0.15	0.29	0.34	0.14
	Mesorhizobium	0.23	0.35	0.31	0.26	0.29	0.38	0.31	0.27	0.35
	Ilumatobacter	0.3	0.44	0.27	0.22	0.25	0.24	0.27	0.25	0.25
	Steroidobacter	0.25	0.38	0.27	0.25	0.26	0.19	0.3	0.32	0.21
	Gp3	0.18	0.35	0.21	0.21	0.26	0.27	0.23	0.31	0.34
	Others	16.28	25.96	19.49	16.76	20.17	21.04	22.59	23.15	24.09
GENERALIST	Gp6	0.24	0.48	0.31	0.28	0.31	0.35	0.48	0.42	0.38
	Veillonella	0.18	0.2	0.21	0.17	0.2	0.16	0.19	0.2	0.18
	Haemophilus	0.16	0.19	0.18	0.17	0.21	0.19	0.19	0.19	0.19
	Streptococcus	0.16	0.16	0.17	0.17	0.14	0.21	0.16	0.15	0.2
	Gaiella	0.12	0.22	0.14	0.14	0.13	0.18	0.21	0.15	0.18
	Solirubrobacter	0.08	0.19	0.09	0.09	0.12	0.16	0.17	0.13	0.2
	Gemmatimonas	0.06	0.18	0.1	0.09	0.09	0.14	0.14	0.15	0.16
	Porphyromonas	0.11	0.13	0.11	0.12	0.12	0.1	0.13	0.11	0.11
	Prevotella	0.1	0.12	0.1	0.1	0.11	0.11	0.11	0.1	0.11
	Nocardioides	0.06	0.15	0.04	0.06	0.07	0.11	0.09	0.08	0.15

Fusobacterium	0.08	0.1	0.1	0.08	0.1	0.09	0.09	0.09	0.08
Pirellula	0.05	0.1	0.07	0.08	0.08	0.09	0.1	0.1	0.08
Gp16	0.05	0.11	0.05	0.07	0.07	0.08	0.1	0.1	0.09
Neisseria	0.08	0.07	0.07	0.07	0.08	0.06	0.08	0.08	0.07
Rothia	0.07	0.07	0.05	0.06	0.08	0.06	0.06	0.08	0.09
Others	3.11	5.79	3.6	3.66	4.2	4.87	5.32	5.25	5.79

Table S4 Properties of the bacterial cooccurrence networks in different groups

Properties of the networks	CK	L	H
Nodes^a	1075	655	595
Edges^b	16100	1418	1229
Average degree distribution^c	29.953	4.33	4.131
Average clustering coefficient^d	0.457	0.324	0.329
Average path length^e	3.917	6.323	6.859
Modularity^f	0.496	0.698	0.742
Modules	17	82	84
max K-Core	35	8	8
The proportion of OPPORTUNIST	52.84%	55.11%	51.26%
The proportion of SPECIALIST	30.05%	26.72%	31.43%
The proportion of GENERALIST	17.11%	18.17%	17.31%

^a Number of AVSs with the Spearman correlation $|r| > 0.8$ and P-value < 0.05 . ^b Number of significant (P-value < 0.05) correlations between nodes. ^c The larger the average distribution, the more complex the network distribution. ^d How nodes were embedded in their neighborhood, and the degree to which nodes tend to cluster together. ^e The capability of the nodes to form highly connected communities. ^f Modularity > 0.4 suggested that the network has a modular structure.

This study investigated the effects of apple waste liquid fertilizer at different concentrations on soil rhizosphere bacterial community structure and assembly processes, demonstrating that high-concentration treatment significantly altered bacterial community assembly from stochastic to deterministic processes, increased specialist abundance, and formed tighter microbial interaction networks, providing new insights for agricultural waste utilization and soil microbial community regulation.

Areas Requiring Improvement:

1. Experimental Section:

- Include detailed nutritional composition analysis of liquid fertilizer
- Add plant growth parameter measurements
- Consider extending observation period (i.e. 1 year) for long-term effects

2. It lacks functional analyses such as:

- Metabolic pathway prediction
- Functional gene abundance
- Enzyme activities
- Nutrient cycling processes
- Specific roles of key bacterial groups in soil improvement
- Plant growth-promoting traits of identified specialists

3. While the current study focuses mainly on microbial community structural changes, it would benefit from the following key additions to make it more comprehensive: First, include basic fertilizer properties (NPK content, pH), key soil enzyme activities (such as urease and phosphatase), and basic plant growth parameters (plant height, biomass); Second, for the microbial aspects, beyond the existing community structure analysis, incorporate functional gene sequencing or prediction analysis, focusing on nutrient cycling (nitrogen and phosphorus cycles) and plant growth promotion (such as IAA synthesis, phosphate solubilization) pathways. These additions would better explain the fertilizer-microbe-plant interaction mechanisms and enhance the study's practical application value.

4. Writing Style:

- Simplify complex sentences
- Improve clarity and conciseness

For example : Lines 27-29: "The rhizosphere bacterial communities treated with a high concentration of liquid fertilizer exhibited increased diversity and broader niche breadth distribution during the late planting period." The reviewer suggested it to two sentences : "High concentration liquid fertilizer treatment altered rhizosphere bacterial communities. These communities showed increased diversity and broader niche breadth distribution in the late planting period."

Dear Editor Philips Akinwole and dear reviewers:

We feel great thanks for the reviewers' professional comments on our article. Those comments are all valuable and very helpful for revising and improving our paper, as well as the importance guiding significance to our research. According to the editor and reviewers' comments, we have made detailed revisions to the manuscript.

We have done comprehensive modifications all over our manuscript and completely re-done the statistical analysis of the data. We have made total revisions to the background, purpose, and significance of the article. We have added the figures, supplemented extra data, and new references to make our results convincing. We also have improved the English of manuscript. The detailed corrections are listed below.

Reviewer #2

(Public repository details (Required)): Sequence data need to be deposited in a publicly available database.

Response: We thank the reviewer for raising this point. We have uploaded the original data of the manuscript to National Center for Biotechnology Information (NCBI), the BioProject ID is PRJNA1138972 (Line 198-200).

Reviewer #2 (Comments for the Author):

Thank you for the opportunity to review the manuscript titled "Responses of rhizosphere bacterial communities with different niche breadths to liquid fertilizer produced from Fuji apple wastes during planting process." This study sought to examine the effects of liquid fertilizer from apple waste on soil rhizosphere microbial community structure and soil properties. The established replicated experimental plots with bok choy that was amended with low and high concentrations of liquid fertilizer. Significant differences were observed in microbial diversity, abundance, and niche breadth across treatments and time but not community assembly. Key

bacterial indicators like *Sphingomonas* were identified. Specialist, generalist, and opportunist species were identified and compared across treatments and time. Overall, this was a very interesting study with important implications for agriculture. Appropriate and interesting statistical approaches were employed to analyze the data. More details need to be provided regarding the methodology and experimental design. I think the discussion also needs to be expanded a bit with additional focus on the ecology and function of individual taxa. There was also not much information on changes in soil properties over time and/or with the addition of fertilizer which seemed to be a key investigation of the study. The writing was a bit difficult to follow in the introduction and discussion sections and could benefit from additional proofreading and editing. Additional, and specific, comments and recommendations are included in the attached annotated pdf.

Response: Thanks for your suggestions to help us to improve this manuscript. Below we made the point-by-point response against the reviewer's specific comments and the annotated pdf included in the attachment:

We asked an expert in English writing to improve our manuscript and added this expert as a co-author (line 5, 9 and 10).

We amended the "e-mail" to "E-mail". (Line 13)

We deleted "as the substrates" in Line 22.

We changed sentence the to "Furthermore, the application mainly increased the niche breadth of bacterial communities, ...". (Line 27-28)

We deleted "breadth distribution" in Line 31.

'Niche breadth' is essentially the range or variety of conditions defining a species' niche, and multi-omics applied to microbial communities offers a great opportunity to observe of realized niche differences at the taxonomic level [1,2]. Therefore, in this manuscript, we continue to use the writing method of "Niche breadth" and calculate the niche breadth of each species in the microbiota through the package of "spa". (start at Line 36)

To make it clearer, we rewrote the sentences to “...the number and relative abundance of specialist ASVs such as *Sphingomonas*, Gp6 and *Arthrobacter* increased”. (Line 39-40)

We modified all the SPECIALIST/GENERALIST/OPPORTUNIST capitalized in the manuscript to lowercase by referring to expressions in references [3,4]. (Line 39, 43, 240-243, 329, 333, 339, 341, 352, 376, 378, 383, 385, 387-388, 390, 398-399, 416, 428-429, 435, 528, 534, 539, 543, 551, 572, 575, 595, 630, 633 and 634)

We modified all the sentence to “The issue of arable soil degradation is of global importance, ...” (Line 53)

We rewrote the sentence to “The bioinformatics analysis of microbiota from field experiments revealed that the application of liquid fertilizer from apple wastes significantly affected the structure and co-occurrence network of rhizosphere bacterial communities. Also, it was observed that the planting time also influenced the community structure.” (Line 57-61)

We rewrote the sentence as “Additionally, the soil available nitrogen (AN) and available phosphorus (AK) were...” (Line 65-66) to make the expression clearer, and corrected the expression of “AN” in Line 201.

We added "an" after "have" (Line 72).

We modified “agro-ecosystem” to “agro-ecosystems”. (Line 73)

We deleted “the” and “the microecosystem of” according to the reviewers’ suggestion. (Line 74 and 75)

We modified the sentence to “..., providing valuable insights regarding their condition”. (Line 82)

We modified “soil” to “soil nutrients”. (Line 89)

We modified “waste applications” to “waste application”. (Line 91)

We modified “employed” to “used”. (Line 96)

We deleted “The” in Line 101 and 105.

We modified the sentence to “Here, we obtained liquid fertilizer from fallen apples through anaerobic fermentation, ...” (Line 116-1117)

We added more accurate GPS coordinates: “latitude 34°21’7’’N, longitude 108°42’34’’E”. (Line 128-129)

We revised the description of randomized block design and replaced Fig.1 with a new Figure to show the experimental design more clearly. (Line 144-148 and Fig.1 in Line 155)

(Line 147 And removed from the plot? Might this significantly disturb the other plants and interconnected rhizosphere?) We referred to relevant studies [5,6] and designed the planting area of each plot to be 25 m² with high planting density, so that we could collect enough samples in different areas during random sampling.

In our study, the same group of samples were used in the same sieve and sterilized at 121 °C for 20 minutes before use. We added the experimental methods to the Line 167-168.

We modified “by the” to “using an”. (Line 174)

We deleted “The” and “measurement”. (Line 175-176)

We modified the version number of Usearch v10 and added related links. (Line 185-186)

We added the version number v11.4 of RDP database. (Line 197)

We modified the sampling method to “Soil samples at the full depth of the profile was evenly mixed, ...” (Line 205)

We added “to” after “prior”. (Line 207)

We added methods to detect pH and conductivity: “...using the FE28 desktop pH and conductivity meter (Mettler Toledo, USA)”. (Line 217-218)

We added “species was” before “based”. (Line 241)

We modified “performed” to “calculated”. (Line 271)

We modified “the” to “fertilizer”. (Line 274)

We modified “concentration” to “concentrations”. (Line 277)

We modified the sentence to “The Bray-Curtis dissimilarity of group H showed a significant decrease...” (Line 286-287)

We modified the layout of Figure 2 and increased the font size to make the annotations in Figure 2B clearer. (Line 294)

We modified the sentence to “It was also observed that high treatment concentrations led to a significant increase in niche breadth, ...” (Line 310-311)

According to relevant references [1,2], we unified the expression of all “niche width” in the manuscript as “niche breadth”. (Fig. 3 in Line 319 and Line 523)

We modified the unclear sentence to “It was observed that the relative abundance of specialist species was higher (from 52.84% to 69.49%), followed by opportunist species (from 25.13% to 38.89%).” (Line 328-330)

We added “fertilizer” before “concentration”. (Line 335)

We added “a” before “co-occurrence network”. (Line 351)

We modified “specialist” to “specialist species”. (Line 376)

We added “cluster” after “opportunist”. (Line 378)

(what about trying a differential abundance analysis between treatments?) In soil studies, dominant bacterial groups directly affected habitat preferences [7]. Therefore, our study focused on the ASVs with the top 30 relative abundance, and we also conducted a differential analysis of the ASVs with the top 30 relative abundance. (Line 427)

(was this characterized for the liquid fertilizer?) The liquid fertilizer (Agricultural Jiaosu, AJ) was produced through an anaerobic fermentation process, which resulted in the growth of Firmicutes, Actinobacteria and many other microorganisms [8]. Therefore, liquid fertilizers do provide beneficial microorganisms. (Line 480)

We modified “remained” to “remains”. (Line 503)

We modified “Oppositely” to “Conversely”. (Line 512)

We added “A” before “previous study”. (Line 548)

We modified the expression of the sentence to “In agro-ecosystems, The microbial diversity could provide a diverse range of ecological

functions and enhance soil multifunctionality.”, and added relevant references [9]. (Line 565-567)

(Need more discussion on why this might be.) We added a discussion on the specialist and generalist species, as well as their association with soil physicochemical properties, and added relevant references: “This difference might arise from variations in multifunctionality among sub-groups, with soil specialist ASVs involved in the mineralization of organic compounds, while generalist ASVs contributed to the release of fixed nutrients [10,11].” (Line 573-576)

Reviewer #3: (Comments for the Author):

This study investigated the effects of apple waste liquid fertilizer at different concentrations on soil rhizosphere bacterial community structure and assembly processes, demonstrating that high-concentration treatment significantly altered bacterial community assembly from stochastic to deterministic processes, increased specialist abundance, and formed tighter microbial interaction networks, providing new insights for agricultural waste utilization and soil microbial community regulation.

Areas Requiring Improvement:

1. Experimental Section:

- Include detailed nutritional composition analysis of liquid fertilizer
- Add plant growth parameter measurements
- Consider extending observation period (i.e. 1 year) for long-term effects

Response: We thank the reviewer for raising this point. Firstly, we provided a detailed description of the fermentation process of apple wastes liquid fertilizer and conduct tests on the key indicators that affect soil ecology: pH, soluble solid content, and total acid. Next, our manuscript mainly focuses on examining the impact of liquid fertilizer on the physical

and chemical properties of soil as well as the soil microenvironment. When diluted with a large volume of water (200 times and 400 times), various components in the liquid fertilizer were significantly reduced. Its impact on soil mainly resulted from stimulating soil microorganisms and seedling growth. Consequently, our study did not perform an extensive analysis specifically targeting the composition of the liquid fertilizer itself, and the strategy consistent with previous studies [8,12].

We added the parameters of plant growth and conducted statistical analysis in the manuscript. “Effects of liquid fertilizer application on plant growth” was added to “Results” part (Line 459-465) and “Liquid fertilizer application significantly promoted the growth of plants” was added to “Discussion” part (Line 612-618). Data collection methods were also added in “Materials and methods” (Line 202 and 219-222).

This manuscript primarily focused on the short-term effects of liquid fertilizer application on soil, and we will continue to conduct long-term research on the effects of liquid fertilizer application on soil in the future.

2. It lacks functional analyses such as:

- Metabolic pathway prediction
- Functional gene abundance
- Enzyme activities
- Nutrient cycling processes
- Specific roles of key bacterial groups in soil improvement
- Plant growth-promoting traits of identified specialists

Response: We thank the reviewer for raising this point. We performed Tax4fun2 package to predict the function of bacterial community and discussed the possible effects of liquid fertilizer on bacterial community function. “Effects of liquid fertilizer application on soil bacterial community function” was added to “Results” part (Line 446-458), and “Liquid fertilizer

application significantly affected the function of soil bacteria community” was added to “Discussion” part (Line 600-611). Functional prediction methods were also added in “Materials and methods” (Line 245-247).

3. While the current study focuses mainly on microbial community structural changes, it would benefit from the following key additions to make it more comprehensive: First, include basic fertilizer properties (NPK content, pH), key soil enzyme activities (such as urease and phosphatase), and basic plant growth parameters (plant height, biomass); Second, for the microbial aspects, beyond the existing community structure analysis, incorporate functional gene sequencing or prediction analysis, focusing on nutrient cycling (nitrogen and phosphorus cycles) and plant growth promotion (such as IAA synthesis, phosphate solubilization) pathways. These additions would better explain the fertilizer-microbe-plant interaction mechanisms and enhance the study's practical application value.

Response: We thank the reviewer for pointing this out. We have added the results of soil bacterial community function to the manuscript, and discussed the effect of liquid fertilizer concentration on community function (Line 446-458 and Line 600-611). At the same time, we also added the growth parameters of bok choy and discussed the promotion effect of liquid fertilizer on plant growth (Line 459-465 and Line 600-611). We added Figure 7 and the corresponding data (tables S5 and S6), and added the contents of the effects of liquid fertilizer on soil function and plant growth in “Abstract” (Line 43-50), “Importance” (Line 67-69) and “Conclusions” (Line 635-637) parts.

According to the reviewers’ comments, we have revised the manuscript extensively. If there are any other modifications we could make, we would like very much to modify them and we really appreciate your help. “Microbiology Spectrum” is a journal of great popularity and prestige. We

hope that our manuscript could be considered for publication in your journal.
Thank you very much for your help.

References

- [1] Sexton, J.P., Montiel, J., Shay, J.E., Stephens, M.R., Slatyer, R.A., 2017. Evolution of Ecological Niche Breadth. *Annu. Rev. Ecol. Evol. Syst.*, 2017, **48**, 183–206.
- [2] Malard, L.A., Guisan, A., 2023. Into the microbial niche. *Trends in Ecology & Evolution*, 2023, **38**, 936–945.
- [3] Abdullah Al, M., Xue, Y., Xiao, P., Xu, J., et al., 2022. Community assembly of microbial habitat generalists and specialists in urban aquatic ecosystems explained more by habitat type than pollution gradient. *Water Research*, 2022, **220**, 118693.
- [4] Gu, Y., Lin, S., Mo, Y., Li, L., et al., 2024. Niche features and assembly mechanisms of microeukaryotic generalists and specialists along a north-south gradient of a subtropical coastal sea. *Mar. Ecol. Prog. Ser.*, 2024, **742**, 35–57.
- [5] Hou, S., Ren, X., Yang, Y., Wang, D., et al., 2022. Genome-Wide Development of Polymorphic Microsatellite Markers and Association Analysis of Major Agronomic Traits in Core Germplasm Resources of Tartary Buckwheat. *Front. Plant Sci.*, 2022, **13**, 819008.
- [6] Wang, Y., Jiang, W., Liu, H., Zeng, Y., et al., 2017. Marker assisted pyramiding of Bph6 and Bph9 into elite restorer line 93–11 and development of functional marker for Bph9. *Rice*, 2017, **10**, 51.
- [7] Delgado-Baquerizo, M., Oliverio, A.M., Brewer, T.E., Benavent-González, A., et al., 2018. A global atlas of the dominant bacteria found in soil. *Science*, 2018, **359**, 320–325.
- [8] Cheng, X., Gao, Y., Wang, Z., Cai, Y., Wang, X., 2023. Agricultural Jiaosu Enhances the Stress Resistance of Pak Choi (*Brassica rapa* L. subsp. *chinensis*) by Recruiting Beneficial Rhizosphere Bacteria and Altering Metabolic Pathways. *Agronomy*, 2023, **13**, 2310.
- [9] Qiu, L., Zhang, Q., Zhu, H., Reich, P.B., et al., 2021. Erosion reduces soil microbial diversity, network complexity and multifunctionality. *The ISME Journal*, 2021, **15**, 2474–2489.
- [10] Bar-Massada, A., 2015. Complex relationships between species niches and environmental heterogeneity affect species co-occurrence patterns in modelled and real communities. *P Roy Soc B-biol Sci*, 2015, **282**, 20150927.
- [11] He, C., Han, T., Tan, L., Li, X., 2022. Effects of Dark Septate Endophytes on the Performance and Soil Microbia of Lycium

ruthenicum Under Drought Stress. *Front. Plant Sci.*, 2022, **13**, 898378.

[12] Li, H., Luo, N., Ji, C., Li, J., et al., 2022. Liquid Organic Fertilizer Amendment Alters Rhizosphere Microbial Community Structure and Co-occurrence Patterns and Improves Sunflower Yield Under Salinity-Alkalinity Stress. *Microb Ecol*, 2022, **84**, 423–438.

Re: Spectrum02068-24R1 (Responses of rhizosphere bacterial communities with different niche breadths to liquid fertilizer produced from Fuji apple wastes during planting process)

Dear Mr. Fan Chang:

Thank you for the privilege of reviewing your work. Below you will find my comments, instructions from the Spectrum editorial office, and the reviewer comments.

I now have comments back (from Reviewer 2) on your revised manuscript and other comments from an additional reviewer.

Revision Guidelines

Sincerely,
Philips Akinwole
Editor
Microbiology Spectrum

Reviewer #2 (Comments for the Author):

Thank you for addressing all of my previous comments. There are a few minor edits needed regarding the methods section (e.g., clarification on the experimental design/sampling and included the predicted functional analysis in the methods) as indicated in the attached pdf.

Reviewer #3 (Comments for the Author):

This study on the impact of liquid fertilizer produced from apple waste on soil microbial diversity and crop growth is of significant academic value and practical importance. However, there are areas that need improvement.

1. Liquid Fertilizer Processing and Physicochemical Parameters

- The article mentions the use of a 200-micron filter for liquid fertilizer, but it does not provide detailed information about the remaining components after filtration, particularly the removal of cellulose and the types and concentrations of remaining organic matter. It is recommended that the authors supplement this information to help readers understand the main active components of the liquid fertilizer.
- The total acid concentration of 1.5 g/L is relatively low. It is suggested that the authors provide more details about the measurement method and specify which organic acids are included. Additionally, the authors should include key parameters such as nitrogen (N), phosphorus (P), potassium (K), and whether humic acids or other organic matter are present. This information is crucial for understanding the fertilizer's efficacy and its impact on soil microbes.

2. Article Structure and Content Organization

- **Crop Growth Impact:** The article dedicates significant space to describing changes in microbial communities but devotes relatively little attention to the impact on crop growth. It is suggested that the authors move the discussion of fertilizer effects on crops to the first section of the results and discussion. This would better highlight the practical applications of the research.
- **Soil Physicochemical Properties:** The short-term changes in soil physicochemical properties caused by the fertilizer could be discussed in the second section, serving as an explanation for how the fertilizer promotes crop growth.
- **Microbial Community Changes:** The analysis of soil microbial community changes, along with statistical analysis and functional predictions, could be placed in the third section. This would allow the authors to explain, from a microbial perspective, why the fertilizer leads to improved outcomes.

3. Deepening Functional Predictions

- While the article discusses changes in KEGG pathways related to microbial functional predictions, it lacks a deeper analysis of how these functional changes specifically influence soil fertility and crop growth. It is recommended that the authors further explore the practical implications of these functional predictions to strengthen the study's logic and persuasiveness.

4. Study Limitations

- The article focuses primarily on bacterial communities, but soil microbial communities also include fungi, archaea, and other organisms. It is suggested that future studies consider a broader range of microbial groups to provide a more comprehensive understanding.
- The liquid fertilizer concentrations tested in this study are relatively simple. Future research could explore a wider range of concentrations to more fully assess their effects on soil and crops.

**Responses of rhizosphere bacterial communities with**
**different niche breadths to liquid fertilizer produced from**
***Fuji* apple wastes during planting process**

Running Title: Effects of liquid fertilizer and planting on soil microbiota

Fengan Jia ¹, Fan Chang ^{1*}, Qingan Jia ², Yan Li ¹, Lisha Zhen ¹, Rui Lv ¹, Yun Xie ³

1 Shaanxi Institute of Microbiology, Xi'an 710043, China

2 Institute of Medical Research, Northwestern Polytechnical University, Xi'an 710072,
China

3 Department of clinical laboratory, Northwest Women's and Children's Hospital,
Xi'an, Shaanxi, China

Mailing address: 76 Xiying Road, Xi'an 710043, China

E-mail: fox387@163.com

Tel.: +8613709259407

[revised manuscript text omitted]

found that high concentration of liquid fertilizer could significantly reduce the
abundance of functional genes of human diseases in soil, including infectious diseases,
endocrine and metabolic diseases, and substance dependence.

Among plant health parameters, plant height was considered a primary indicator
of plant growth rate [77]. Our study found that liquid fertilizer treatment could
significantly increase the plant height of bok choy, and this effect was
concentration-dependent. Our study also confirmed that the dry weight of bok choy
increased significantly after applying liquid fertilizer, indicating that secondary
metabolites in liquid fertilizer and soil microorganisms might work together to
improve the biomass of bok choy [78,79].

**Conclusions**

This study revealed the succession of bacterial communities following different
concentrations of apple waste liquid fertilizer treatment during planting, while also
highlighting the impact of bacteria communities on soil properties. The Chao1 index,
phylogenetic diversity index, beta diversity distance, and Bray-Curtis dissimilarity of

bacteria exhibited significant differences at high concentrations, especially during the
later stages of planting. The concentration of liquid fertilizer was also found to have a
significant impact on the distribution of niche breadth among rhizosphere bacteria.
The rhizosphere bacterial communities exhibited reduced network complexity,
including a decrease in the number of nodes, edges, and overall network size
following the application of liquid fertilizer. The number of central specialist
indicators of high-abundance ASVs was increased following the use of liquid fertilizer.
In particular, *Sphingomonas* were identified as hubs and connectors within the key
indicators in the groups where liquid fertilizer was applied. The study also found that
specialist ASVs was significantly correlated with AP, while generalist ASVs was
significantly correlated with AN. Furthermore, liquid fertilizer treatment significantly
affected the metabolism and genetic information processing of soil bacterial
community, and significantly promoted the growth of plants. This study had expanded
our understanding of the interaction between bacterial communities and soil
properties during planting with varying concentrations of apple wastes liquid fertilizer,
thereby providing new insights into bacteria regulation of rhizosphere microhabitats.

**Funding**

This study was supported by Science and Technology Project of Shaanxi
Academy of Science (2024k-11).

**Reference**

- [1] Philippot, L., Chenu, C., Kappler, A., Rillig, M.C., Fierer, N., 2024.
The interplay between microbial communities and soil properties. *Nat Rev*
*Microbiol*, 2024, **22**, 226–239.
- [2] Peng, Z., Qian, X., Liu, Y., Li, X., et al., 2024. Land conversion to
agriculture induces taxonomic homogenization of soil microbial communities
globally. *Nat Commun*, 2024, **15**, 3624.
- [3] Jing, J., Cong, W.-F., Bezemer, T.M., 2022. Legacies at work: plant–
soil–microbiome interactions underpinning agricultural sustainability. *Trends*
*Plant Sci*, 2022, **27**, 781–792.

- [4] Wu, H., Cui, H., Fu, C., Li, R., et al., 2024. Unveiling the crucial
role of soil microorganisms in carbon cycling: A review. *Science of The Total*
*Environment*, 2024, **909**, 168627.
- [5] Hartmann, M., Six, J., 2022. Soil structure and microbiome
functions in agroecosystems. *Nat Rev Earth Env*, 2022, **4**, 4–18.
- [6] El-Saadony, M.T., Saad, A.M., Soliman, S.M., Salem, H.M., et al.,
2022. Plant growth-promoting microorganisms as biocontrol agents of plant
diseases: Mechanisms, challenges and future perspectives. *Front Plant Sci*, 2022,
**13**, 923880.
- [7] Shi, G., Sun, H., Calderón-Urrea, A., Li, M., et al., 2021. Bacterial
communities as indicators of soil health under a continuous cropping system.
*Land Degrad Dev*, 2021, **32**, 2393–2408.
- [8] Fierer, N., Wood, S.A., Bueno de Mesquita, C.P., 2021. How
microbes can, and cannot, be used to assess soil health. *Soil Biology and*
*Biochemistry*, 2021, **153**, 108111.
- [9] Thepbandit, W., Athinuwat, D., 2024. Rhizosphere Microorganisms
Supply Availability of Soil Nutrients and Induce Plant Defense. *Microorganisms*,
2024, **12**, 558.
- [10] Schmidt, J.E., Vannette, R.L., Igwe, A., Blundell, R., et al., 2019.
Effects of Agricultural Management on Rhizosphere Microbial Structure and
Function in Processing Tomato Plants. *Appl Environ Microb*, 2019, **85**,
e01064-19.
- [11] Wang, T., Yang, K., Ma, Q., Jiang, X., et al., 2022. Rhizosphere
Microbial Community Diversity and Function Analysis of Cut Chrysanthemum
During Continuous Monocropping. *Front Microbiol*, 2022, **13**, 801546.
- [12] Toor, G.S., Yang, Y.-Y., Das, S., Dorsey, S., Felton, G., 2021.
*Advances in Agronomy*, vol. 168, Elsevier, pp. 157–201.
- [13] De Corato, U., 2020. Agricultural waste recycling in horticultural
intensive farming systems by on-farm composting and compost-based tea
application improves soil quality and plant health: A review under the
perspective of a circular economy. *Science of The Total Environment*, 2020, **738**,
139840.
- [14] Liu, J., Shu, A., Song, W., Shi, W., et al., 2021. Long-term organic
fertilizer substitution increases rice yield by improving soil properties and
regulating soil bacteria. *Geoderma*, 2021, **404**, 115287.
- [15] Garbowski, T., Bar-Michalczyk, D., Charazińska, S.,
Grabowska-Polanowska, B., et al., 2023. An overview of natural soil
amendments in agriculture. *Soil and Tillage Research*, 2023, **225**, 105462.
- [16] Elnahal, A.S.M., El-Saadony, M.T., Saad, A.M., Desoky, E.-S.M., et
al., 2022. The use of microbial inoculants for biological control, plant growth
promotion, and sustainable agriculture: A review. *Eur J Plant Pathol*, 2022, **162**,
759–792.

- [17] Mengqi, Z., Shi, A., Ajmal, M., Ye, L., Awais, M., 2023.
Comprehensive review on agricultural waste utilization and high-temperature
fermentation and composting. *Biomass Convers Bior*, 2023, **13**, 5445–5468.
- [18] Martín-Lammerding, D., Gabriel, J.L., Zambrana, E.,
Santín-Montanyá, I., Tenorio, J.L., 2021. Organic Amendment vs. Mineral
Fertilization under Minimum Tillage: Changes in Soil Nutrients, Soil Organic
Matter, Biological Properties and Yield after 10 Years. *Agriculture*, 2021, **11**,
700.
- [19] Dahunsi, S.O., Oranusi, S., Efeovbokhan, V.E., Adesulu-Dahunsi,
699 A.T., Ogunwole, J.O., 2021. Crop performance and soil fertility improvement
using organic fertilizer produced from valorization of Carica papaya fruit peel.
*Sci Rep-uk*, 2021, **11**, 4696.
- [20] Bello, S.K., Alayafi, A.H., AL-Solaimani, S.G., Abo-Elyousr,
703 K.A.M., 2021. Mitigating Soil Salinity Stress with Gypsum and Bio-Organic
Amendments: A Review. *Agronomy*, 2021, **11**, 1735.
- [21] Anyaoha, K.E., Sakrabani, R., Patchigolla, K., Mouazen, A.M., 2018.
Critical evaluation of oil palm fresh fruit bunch solid wastes as soil amendments:
Prospects and challenges. *Resour Conserv Recy*, 2018, **136**, 399–409.
- [22] Ramamoorthy, K., Dhanraj, R., Vijayakumar, N., Ma, Y., et al., 2024.
Vegetable and fruit wastes: Valuable source for organic fertilizer for effective
growth of short-term crops: *Solanum lycopersicum* and *Capsicum annum*.
*Environ Res*, 2024, **251**, 118727.
- [23] Puspitawati, M.D., Sumiasih, I.H., 2021. Organic Fertilizer from
Starfruit Waste Sustainable Agriculture Solution. *IOP Conf. Ser.: Earth Environ.*
*Sci.*, 2021, **709**, 012069.
- [24] Macias-Benitez, S., Garcia-Martinez, A.M., Caballero Jimenez, P.,
Gonzalez, J.M., et al., 2020. Rhizospheric Organic Acids as Biostimulants:
Monitoring Feedbacks on Soil Microorganisms and Biochemical Properties.
*Front Plant Sci*, 2020, **11**, 633.
- [25] Becker, W., Helsing, E. (Eds.), 1991. Food and health data: their use
in nutrition policy-making., World Health Organization, Regional Office for
Europe, Copenhagen.
- [26] Logue, J.B., Stedmon, C.A., Kellerman, A.M., Nielsen, N.J., et al.,
2016. Experimental insights into the importance of aquatic bacterial community
composition to the degradation of dissolved organic matter. *ISME J*, 2016, **10**,
533–545.
- [27] Edgar, R.C., 2010. Search and clustering orders of magnitude faster
than BLAST. *Bioinformatics*, 2010, **26**, 2460–2461.
- [28] Edgar, R.C., 2018. Updating the 97% identity threshold for 16S
ribosomal RNA OTUs. *Bioinformatics*, 2018, **34**, 2371–2375.
- [29] Knight, R., Vrbanac, A., Taylor, B.C., Aksenov, A., et al., 2018. Best
practices for analysing microbiomes. *Nature Reviews Microbiology*, 2018, **16**,

410–422.

[30] Bacci, G., Bani, A., Bazzicalupo, M., Ceccherini, M.T., et al., 2015.

Evaluation of the Performances of Ribosomal Database Project (RDP) Classifier

for Taxonomic Assignment of 16S rRNA Metabarcoding Sequences Generated

from Illumina-Solexa NGS. *J. Genomics*, 2015, **3**, 36–39.

[31] Qu, B., Liu, Y., Sun, X., Li, S., et al., 2019. Effect of various

mulches on soil physico—Chemical properties and tree growth (*Sophora*

*japonica*) in urban tree pits. *PLoS ONE*, 2019, **14**, e0210777.

[32] Bradstreet, R.B., 1954. Kjeldahl Method for Organic Nitrogen.

*Analytical Chemistry*, 1954, **26**, 185–187.

[33] Khan, S.A., Mulvaney, R.L., Mulvaney, C.S., 1997. Accelerated

Diffusion Methods for Inorganic-Nitrogen Analysis of Soil Extracts and Water.

*Soil Science Society of America Journal*, 1997, **61**, 936–942.

[34] Murphy, J., Riley, J.P., 1962. A modified single solution method for

the determination of phosphate in natural waters. *Analytica Chimica Acta*, 1962,

**27**, 31–36.

[35] Wolf, A.M., Baker, D.E., 1985. Comparisons of soil test phosphorus

by Olsen, Bray P1, Mehlich I and Mehlich III methods. *Communications in Soil*

*Science and Plant Analysis*, 1985, **16**, 467–484.

[36] Attoe, O.J., Truog, E., 1947. Rapid Photometric Determination of

Exchangeable Potassium and Sodium. *Soil Science Society of America Journal*,

1947, **11**, 221–226.

[37] Fan, K., Weisenhorn, P., Gilbert, J.A., Shi, Y., et al., 2018. Soil pH

correlates with the co-occurrence and assemblage process of diazotrophic

communities in rhizosphere and bulk soils of wheat fields. *Soil Biology and*

*Biochemistry*, 2018, **121**, 185–192.

[38] Chen, T., Liu, Y., Huang, L., 2022. ImageGP: An easy-to-use data

visualization web server for scientific researchers. *iMeta*, 2022, **1**.

[39] Oksanen, J., Blanchet, F.G., Friendly, M., Kindt, R., et al., 2019.

*vegan: Community Ecology Package*.

[40] Kembel, S.W., Cowan, P.D., Helmus, M.R., Cornwell, W.K., et al.,

2010. Picante: R tools for integrating phylogenies and ecology. *Bioinformatics*,

2010, **26**, 1463–1464.

[41] Stegen, J.C., Lin, X., Konopka, A.E., Fredrickson, J.K., 2012.

Stochastic and deterministic assembly processes in subsurface microbial

communities. *ISME J*, 2012, **6**, 1653–1664.

[42] Levins, R., 1968. Evolution in changing environments: some

theoretical explorations. *Monographs in Population Biology*, 1968.

[43] Wu, W., Lu, H.-P., Sastri, A., Yeh, Y.-C., et al., 2018. Contrasting the

relative importance of species sorting and dispersal limitation in shaping marine

bacterial versus protist communities. *The ISME Journal*, 2018, **12**, 485–494.

[44] Wemheuer, F., Taylor, J.A., Daniel, R., Johnston, E., et al., 2020.

Tax4Fun2: prediction of habitat-specific functional profiles and functional
redundancy based on 16S rRNA gene sequences. *Environmental Microbiome*,
2020, **15**, 11.

[45] Cáceres, M.D., Legendre, P., 2009. Associations between species
and groups of sites: indices and statistical inference. *Ecology*, 2009, **90**, 3566–
3574.

[46] Bastian, M., Heymann, S., Jacomy, M., 2009. Gephi: An Open
Source Software for Exploring and Manipulating Networks. *ICWSM*, 2009, **3**,
361–362.

[47] Diniz-Filho, J.A.F., Soares, T.N., Lima, J.S., Dobrovolski, R., et al.,
2013. Mantel test in population genetics. *Genet Mol Biol.*, 2013, **36**, 475–485.

[48] Gao, Y., Zhang, Y., Cheng, X., Zheng, Z., et al., 2022. Agricultural
Jiaosu: An Eco-Friendly and Cost-Effective Control Strategy for Suppressing
*Fusarium Root Rot Disease in Astragalus membranaceus*. *Front Microbiol*, 2022,
**13**, 823704.

[49] Cheng, X., Gao, Y., Wang, Z., Cai, Y., Wang, X., 2023. Agricultural
Jiaosu Enhances the Stress Resistance of Pak Choi (*Brassica rapa* L. subsp.
*chinensis*) by Recruiting Beneficial Rhizosphere Bacteria and Altering Metabolic
Pathways. *Agronomy*, 2023, **13**, 2310.

[50] Coller, E., Oliveira Longa, C.M., Morelli, R., Zanoni, S., et al., 2021.
Soil Communities: Who Responds and How Quickly to a Change in Agricultural
System? *Sustainability-basel*, 2021, **14**, 383.

[51] Beschoren da Costa, P., Benucci, G.M.N., Chou, M.-Y., Van Wyk, J.,
et al., 2022. Soil Origin and Plant Genotype Modulate Switchgrass Aboveground
Productivity and Root Microbiome Assembly. *Mbio*, 2022, **13**, e00079-22.

[52] Barnett, S.E., Youngblut, N.D., Buckley, D.H., 2020. Soil
characteristics and land-use drive bacterial community assembly patterns. *FEMS*
*Microbiology Ecology*, 2020, **96**, fiz194.

[53] Zhou, X., Khashi u Rahman, M., Liu, J., Wu, F., 2021. Soil
acidification mediates changes in soil bacterial community assembly processes
in response to agricultural intensification. *Environ Microbiol*, 2021, **23**, 4741–
4755.

[54] Xu, Q., Vandenkoornhuysse, P., Li, L., Guo, J., et al., 2022. Microbial
generalists and specialists differently contribute to the community diversity in
farmland soils. *Journal of Advanced Research*, 2022, **40**, 17–27.

[55] Baquero, F., Coque, T.M., Galán, J.C., Martinez, J.L., 2021. The
Origin of Niches and Species in the Bacterial World. *Front Microbiol*, 2021, **12**,
657986.

[56] Hernandez, D.J., Kieseewetter, K.N., Almeida, B.K., Revillini, D.,
Afkhami, M.E., 2023. Multidimensional specialization and generalization are
pervasive in soil prokaryotes. *Nat Ecol Evol*, 2023, **7**, 1408–1418.

[57] Gao, M., Xiong, C., Gao, C., Tsui, C.K.M., et al., 2021.

Disease-induced changes in plant microbiome assembly and functional
adaptation. *Microbiome*, 2021, **9**, 187.

[58] Compant, S., Clément, C., Sessitsch, A., 2010. Plant
growth-promoting bacteria in the rhizo- and endosphere of plants: Their role,
colonization, mechanisms involved and prospects for utilization. *Soil Biology
and Biochemistry*, 2010, **42**, 669–678.

[59] Bar-Massada, A., 2015. Complex relationships between species
niches and environmental heterogeneity affect species co-occurrence patterns in
modelled and real communities. *P Roy Soc B-biol Sci*, 2015, **282**, 20150927.

[60] Liu, C., Li, X., Mansoldo, F.R.P., An, J., et al., 2022. Microbial
habitat specificity largely affects microbial co-occurrence patterns and functional
profiles in wetland soils. *Geoderma*, 2022, **418**, 115866.

[61] Cardinale, M., Ratering, S., Sadeghi, A., Pokhrel, S., et al., 2020.
The Response of the Soil Microbiota to Long-Term Mineral and Organic
Nitrogen Fertilization is Stronger in the Bulk Soil than in the Rhizosphere.
*Genes-basel*, 2020, **11**, 456.

[62] Ye, Z., Wang, J., Li, J., Liu, G., et al., 2022. Different roles of core
and noncore bacterial taxa in maintaining soil multinutrient cycling and
microbial network stability in arid fertigation agroecosystems. *J Appl Ecol*, 2022,
**59**, 2154–2165.

[63] Ishimoto, C.K., Aono, A.H., Nagai, J.S., Sousa, H., et al., 2021.
Microbial co-occurrence network and its key microorganisms in soil with
permanent application of composted tannery sludge. *Science of The Total
Environment*, 2021, **789**, 147945.

[64] Guo, Y., Song, B., Li, A., Wu, Q., et al., 2022. Higher pH is
associated with enhanced co-occurrence network complexity, stability and
nutrient cycling functions in the rice rhizosphere microbiome. *Environmental
Microbiology*, 2022, **24**, 6200–6219.

[65] Lin, Q., Li, L., Adams, J.M., Heděnc, P., et al., 2021. Nutrient
resource availability mediates niche differentiation and temporal co-occurrence
of soil bacterial communities. *Appl Soil Ecol*, 2021, **163**, 103965.

[66] Qiu, L., Zhang, Q., Zhu, H., Reich, P.B., et al., 2021. Erosion
reduces soil microbial diversity, network complexity and multifunctionality. *The
ISME Journal*, 2021, **15**, 2474–2489.

[67] Jiao, S., Yang, Y., Xu, Y., Zhang, J., Lu, Y., 2020. Balance between
community assembly processes mediates species coexistence in agricultural soil
microbiomes across eastern China. *ISME J*, 2020, **14**, 202–216.

[68] Favela, A., O. Bohn, M., D. Kent, A., 2021. Maize germplasm
chronosequence shows crop breeding history impacts recruitment of the
rhizosphere microbiome. *ISME J*, 2021, **15**, 2454–2464.

[69] He, C., Han, T., Tan, L., Li, X., 2022. Effects of Dark Septate
Endophytes on the Performance and Soil Microbia of *Lycium ruthenicum* Under

Drought Stress. *Front. Plant Sci.*, 2022, **13**, 898378.

[70] Bar-Massada, A., 2015. Complex relationships between species
niches and environmental heterogeneity affect species co-occurrence patterns in
modelled and real communities. *P Roy Soc B-biol Sci*, 2015, **282**, 20150927.

[71] Sireci, M., Muñoz, M.A., Grilli, J., 2023. Environmental
fluctuations explain the universal decay of species-abundance correlations with
phylogenetic distance. *Proc. Natl. Acad. Sci. U.S.A.*, 2023, **120**, e2217144120.

[72] Asaf, S., Numan, M., Khan, A.L., Al-Harrasi, A., 2020.
*Sphingomonas* : from diversity and genomics to functional role in environmental
remediation and plant growth. *Crit Rev Biotechnol*, 2020, **40**, 138–152.

[73] Liu, S., Liu, R., Zhang, S., Shen, Q., et al., 2024. The Contributions
of Sub-Communities to the Assembly Process and Ecological Mechanisms of
Bacterial Communities along the Cotton Soil–Root Continuum Niche Gradient.
*Microorganisms*, 2024, **12**, 869.

[74] Ge, Z., Li, S., Bol, R., Zhu, P., et al., 2021. Differential long-term
fertilization alters residue-derived labile organic carbon fractions and microbial
community during straw residue decomposition. *Soil and Tillage Research*, 2021,
**213**, 105120.

[75] Nardi, S., Muscolo, A., Vaccaro, S., Baiano, S., et al., 2007.
Relationship between molecular characteristics of soil humic fractions and
glycolytic pathway and krebs cycle in maize seedlings. *Soil Biology and*
*Biochemistry*, 2007, **39**, 3138–3146.

[76] Samaddar, S., Karp, D.S., Schmidt, R., Devarajan, N., et al., 2021.
Role of soil in the regulation of human and plant pathogens: soils’ contributions
to people. *Phil. Trans. R. Soc. B*, 2021, **376**, 20200179.

[77] Khan, H., Kaur, S., Baldwin, T.C., Radecka, I., et al., 2020. Effective
Control against Broadleaf Weed Species Provided by Biodegradable PBAT/PLA
Mulch Film Embedded with the Herbicide 2-Methyl-4-Chlorophenoxyacetic
Acid (MCPA). *ACS Sustainable Chem. Eng.*, 2020, **8**, 5360–5370.

[78] Kang, S.-M., Shaffique, S., Kim, L.-R., Kwon, E.-H., et al., 2021.
Effects of Organic Fertilizer Mixed with Food Waste Dry Powder on the Growth
of Chinese Cabbage Seedlings. *Environments*, 2021, **8**, 86.

[79] Kano, K., Kitazawa, H., Suzuki, K., Widiastuti, A., et al., 2021.
Effects of Organic Fertilizer on Bok Choy Growth and Quality in Hydroponic
Cultures. *Agronomy*, 2021, **11**, 491.

A**B**

A

PageRanks Eccentricity

**CK**
Nodes: 1075
Edges: 16100
Modularity: 0.496
Max K-Core: 35
Positive: 15960 (99.1%)
Negative: 140 (0.9%)

B

PageRanks Eccentricity

**L**
Nodes: 655
Edges: 1418
Modularity: 0.698
Max K-Core: 8
Positive: 1332 (93.9%)
Negative: 86 (6.1%)

C

PageRanks Eccentricity

**H**
Nodes: 595
Edges: 1229
Modularity: 0.742
Max K-Core: 8
Positive: 1145 (93.2%)
Negative: 84 (6.8%)

ASVs and their corresponding genera

ASVs and their corresponding genera

ASVs and their corresponding genera

A

C

B

D

A

KEGG Orthology (level 2)

B

Dear Editor Philips Akinwole and dear reviewers:

We feel great thanks for the reviewers' professional comments on our article. According to the editor and reviewers' comments, we have made detailed revisions to the manuscript. We have done comprehensive modifications all over our manuscript. We have modified the figures, supplemented extra data, and new references to make our results convincing. Finally, we have corrected some spelling mistakes. The detailed corrections are listed below.

Reviewer #2

Thank you for addressing all of my previous comments. There are a few minor edits needed regarding the methods section (e.g., clarification on the experimental design/sampling and included the predicted functional analysis in the methods) as indicated in the attached pdf.

Response: Thanks for your suggestions to help us to improve this manuscript. Below we made the point-by-point response against the reviewer's specific comments and the annotated pdf included in the attachment:

We modified the diagram Figure 1 to detailed chart and modified the description of the corresponding paragraph (line 138, 139,140 and Fig. 1 in line 145).

The Tax4Fun2 method is in line 235-237, we have marked it in red and corrected the spelling mistake.

We fixed the spelling mistake in line 533, add "the" in line 537, and deleted "the" in line 540.

We modified "had found" to "showed". (line 543)

We deleted the extra space in line 548.

We modified "in the specialist" to "in the specialist group". (line 564)

We modified "The microbial diversity could provide a diverse range of ecological functions and enhance soil multifunctionality." to "microbial

communities can provide a wide range of ecological functions and enhance soil multifunctionality.” (line 470-471)

We modified “It was observed a high relative abundance of *Sphingomonas* (ASV_22, ASV_37, ASV_14, and ASV_20), Gp6 (ASV_168, ASV_241, and ASV_264), *Arthrobacter* (ASV_15), with ASV_37, ASV_14, ASV_264, ASV_15 showing a significant increase in group H.” to “*Sphingomonas* (ASV_22, ASV_37, ASV_14, and ASV_20), Gp6 (ASV_168, ASV_241, and ASV_264), and *Arthrobacter* (ASV_15) had high relative abundances across all samples, with ASV_37, ASV_14, ASV_264, ASV_15 showing a significant increase in group H (Fig. 6C and D).” (line 486-489)

We added “in the” before “high-concentration group”. (line 496)

Reviewer #3 (Comments for the Author):

This study on the impact of liquid fertilizer produced from apple waste on soil microbial diversity and crop growth is of significant academic value and practical importance. However, there are areas that need improvement.

1. Liquid Fertilizer Processing and Physicochemical Parameters

- The article mentions the use of a 200-micron filter for liquid fertilizer, but it does not provide detailed information about the remaining components after filtration, particularly the removal of cellulose and the types and concentrations of remaining organic matter. It is recommended that the authors supplement this information to help readers understand the main active components of the liquid fertilizer.

Response: We thank the reviewer for raising this point. In our study, filtration was used to remove the insoluble solids, leaving the liquid phase part, which could be more convenient for dilution and application in subsequent experiments. The main components of liquid fertilizer after fermentation and filtration were soluble solid and total acid, and the relevant components had been written in the "Methods" section: “The liquid fertilizer

had a pH of 4.5, soluble solids of 5.5°Bx, total acid concentration of 3.5 g/L (including 2.4 g/L malic acid).” (line 131-133). This study mainly focused on the effects of the overall liquid fertilizer on soil physicochemical properties, soil bacterial communities and plant growth, and did not involve the effects of specific components of liquid fertilizer on soil. In the future, we will continue to pay attention to the impact of various components in fermentation liquid fertilizer on soil ecology.

- The total acid concentration of 1.5 g/L is relatively low. It is suggested that the authors provide more details about the measurement method and specify which organic acids are included. Additionally, the authors should include key parameters such as nitrogen (N), phosphorus (P), potassium (K), and whether humic acids or other organic matter are present. This information is crucial for understanding the fertilizer's efficacy and its impact on soil microbes.

Response: We thank the reviewer for pointing this out. We rechecked the original data and found that the 3.5 g/L of total acid was incorrectly written as 1.5 g/L, and we have modified it. At the same time, we added the malic acid content (2.4 g/L). (line 132) The content of nitrogen, phosphorus, potassium and other elements in fruits was low, and these nutrients mainly come from the soil during the planting process. Our study mainly focused on the physiological and biochemical characteristics of soil, and we had added the table and related descriptions of physiological and biochemical properties of soil in the supplementary table S7 and line 399.

2. Article Structure and Content Organization

- **Crop Growth Impact:** The article dedicates significant space to describing changes in microbial communities but devotes relatively little attention to the impact on crop growth. It is suggested that the authors move the discussion of fertilizer effects on crops to the first section of the results

and discussion. This would better highlight the practical applications of the research.

Response: We thank the reviewer for pointing this out. We moved the discussion of fertilizer effects on crops to the first section of the Discussion. (line 486-495)

- **Soil Physicochemical Properties:** The short-term changes in soil physicochemical properties caused by the fertilizer could be discussed in the second section, serving as an explanation for how the fertilizer promotes crop growth.

Response: We thank the reviewer for raising this point. We moved our discussion of the short-term changes in the physical and chemical properties of soils caused by fertilizers to in the second section of Discussion. (line 496-529)

- **Microbial Community Changes:** The analysis of soil microbial community changes, along with statistical analysis and functional predictions, could be placed in the third section. This would allow the authors to explain, from a microbial perspective, why the fertilizer leads to improved outcomes.

Response: We thank the reviewer for raising this point. We moved the analysis of soil microbial community changes, as well as statistical analysis and functional prediction, to the end of the discussion. (line 502-653)

3. Deepening Functional Predictions

- While the article discusses changes in KEGG pathways related to microbial functional predictions, it lacks a deeper analysis of how these functional changes specifically influence soil fertility and crop growth. It is recommended that the authors further explore the practical implications of

these functional predictions to strengthen the study's logic and persuasiveness.

Response: We thank reviewer for the comments. We add content on the most abundant functional pathways involved in metabolism (L1) and discuss the possible effects of microbial metabolic processes on soil. (line 641-653)

4. Study Limitations

- The article focuses primarily on bacterial communities, but soil microbial communities also include fungi, archaea, and other organisms. It is suggested that future studies consider a broader range of microbial groups to provide a more comprehensive understanding.

Response: We thank reviewer for the comments. We added limitations to the study of bacterial communities in the “Conclusion” section. (line 680-683)

- The liquid fertilizer concentrations tested in this study are relatively simple. Future research could explore a wider range of concentrations to more fully assess their effects on soil and crops.

Response: We thank the reviewer for raising this point. We added the limitations of low concentration liquid fertilizers and short-term studies to the “Conclusion” section. (line 683-685)

According to the reviewers' comments, we have revised the manuscript extensively. If there are any other modifications we could make, we would like very much to modify them and we really appreciate your help. “Microbiology Spectrum” is a journal of great popularity and prestige. We hope that our manuscript could be considered for publication in your journal. Thank you very much for your help.

Re: Spectrum02068-24R2 (Responses of rhizosphere bacterial communities with different niche breadths to liquid fertilizer produced from Fuji apple wastes during planting process)

Dear Mr. Fan Chang:

Thank you for the privilege of reviewing your work. Below you will find my comments and instructions from the Spectrum editorial office.

Thank you for your detailed response to the reviewer's comments.

In my review, I found some minor edits to attend to in order to improve on the quality and readability of your manuscript. Thank you for your patience.

Minor edits:

In the abstract, AN and AP need to be defined first before abbreviations. Also, the available phosphorus (AK) in the "Importance" section needs to be clarified if it refers to AP in the abstract. Also, see lines 210, 486, etc.

Line 81: Effective microorganisms, plant growth ..., replace effective with "Appropriate..."

Line 158-9: "...and the soil adhered and attached to the roots was collected one rhizosphere soil sample..." should be clarified for grammatical structure.

Include the abbreviations of the samples/category names in the figure's title for easy readability (e.g. CK, L, H).

Lines 410-412 "The results showed that indicators were significantly correlated with soil AN and AP. Additionally, the specialist and generalist subgroups exhibited stronger correlations with soil AN and AP..."

What are the indicators being referred to? include (e.g. and mention the indicators). Also, when using the word stronger, it has to be in comparison to something else. Check "Additionally, the specialist and generalist subgroups exhibited stronger correlations with soil AN and AP..." for grammatical structure.

Lines 457-49: Where specific significant differences in morphological traits are seen, they should be highlighted, given that this was first discussed in the discussion section.

The conclusion that "Our study found that high concentration of liquid fertilizer could significantly reduce the abundance of functional genes of human diseases in soil, including infectious diseases, endocrine and metabolic diseases, and substance dependence" is not sufficiently supported in this study and should be rephrased, e.g. What specific data/figures provided such correlation/causation in this study? A more direct study would be needed to support such an assertion.

Revision Guidelines

Sincerely,
Philips Akinwale
Editor
Microbiology Spectrum

Dear Editor Philips Akinwole and dear reviewers:

We feel great thanks for the reviewers' professional comments on our article. According to the editor comments, we have made revisions to the manuscript. The detailed corrections are listed below.

Minor edits:

In the abstract, AN and AP need to be defined first before abbreviations. Also, the available phosphorus (AK) in the "Importance" section needs to be clarified if it refers to AP in the abstract. Also, see lines 210, 486, etc.

Response: We thank the editor for raising this point. We added the definitions of AN and AP in the "Abstract" (Line 36-37).

The incorrect abbreviation of available phosphorus in "Importance" has been corrected (Line 61).

We made the description of the plant traits in row 210 consistent with the modification in Figure 7 (Line 210-212).

The parts that were not clearly described in Figures 6C and D have been added to correspond to the content of line 486 (Line 424-429).

Line 81: Effective microorganisms, plant growth ..., replace effective with "Appropriate..."

Response: We modified "Effective microorganisms" to "Appropriate microorganisms" in Line 88.

Line 158-9: "...and the soil adhered and attached to the roots was collected one rhizosphere soil sample..." should be clarified for grammatical structure.

Response: We modified the sentence to "... , and the soil adhered to the roots was collected as one rhizosphere soil sample" (Line 154-155).

Include the abbreviations of the samples/category names in the figure's title for easy readability (e.g. CK, L, H).

Response: We add the annotations of the groups in the figures (Line 291-292, Line 313-314, Line 335-336, Line 392-393, Line 434-435, Line 462).

Lines 410-412 "The results showed that indicators were significantly correlated with soil AN and AP. Additionally, the specialist and generalist subgroups exhibited stronger correlations with soil AN and AP..."

What are the indicators being referred to? include (e.g. and mention the indicators). Also, when using the word stronger, it has to be in comparison to something else. Check "Additionally, the specialist and generalist subgroups exhibited stronger correlations with soil AN and AP..." for grammatical structure.

Response: We thank the editor for raising this point. The indicator species were calculated through the *multipatt* function of the 'Indicspecies' package (version 1.7.9), which has been mentioned in the "Methods" section in Line 241-242. We corrected the errors in the format in Line 241.

We modified the sentence to "compared to the opportunist subgroup, the specialist subgroup and the generalist subgroup exhibited a stronger correlation with AN and AP in soil." In Line 406-408.

Lines 457-49: Where specific significant differences in morphological traits are seen, they should be highlighted, given that this was first discussed in the discussion section.

Response: We thank the editor for raising this point. We added the specific descriptions of the differences in morphological trait in Line 457-464.

The conclusion that "Our study found that high concentration of liquid fertilizer could significantly reduce the abundance of functional genes of human diseases in soil, including infectious diseases, endocrine and metabolic diseases, and substance dependence" is not sufficiently supported in this study and should be rephrased, e.g. What specific data/figures provided such correlation/causation in this study? A more direct study would be needed to support such an assertion.

Response: We thank the editor for raising this point. We mentioned this result in Line 450-453, together with the data presented in Figure 7 and Table S5. We corrected the grammatical error in Line 452 and modified the statement in the discussion in Line 616-619 to make the expression clearer: "The KEGG functional prediction results revealed that high concentration of liquid fertilizer significantly reduced the abundance of human disease functional gene pathways in the soil, including infectious diseases: viral, endocrine and metabolic diseases, and substance dependence."

According to the editor's comments, we have revised the manuscript extensively. If there are any other modifications we could make, we would like very much to modify them and we really appreciate your help. "Microbiology Spectrum" is a journal of great popularity and prestige. We hope that our manuscript could be considered for publication in your journal. Thank you very much for your help.

Re: Spectrum02068-24R3 (Responses of rhizosphere bacterial communities with different niche breadths to liquid fertilizer produced from Fuji apple wastes during planting process)

Dear Mr. Fan Chang:

Your manuscript has been accepted, and I am forwarding it to the ASM production staff for publication. Your paper will first be checked to make sure all elements meet the technical requirements. ASM staff will contact you if anything needs to be revised before copyediting and production can begin. Otherwise, you will be notified when your proofs are ready to be viewed.

Sincerely,
Philips Akinwale
Editor
Microbiology Spectrum